# Learning Provably Correct Distributed Protocols Without Human Knowledge

## Abstract

Provably correct distributed protocols, which are a critical component of modern distributed systems, are highly challenging to design and have often required decades of human effort. These protocols allow multiple agents to coordinate to come to a common agreement in an environment with uncertainty and failures, e.g., agents only observe their own state, messages can be lost, and agents may crash. We formulate protocol design as a search problem over strategies in a game with imperfect information, and the desired correctness conditions are specified in Satisfiability Modulo Theories (SMT). However, standard methods for solving multi-agent games fail to learn correct protocols in this setting, even when the number of agents is small. We propose a learning framework, GGMS, which integrates a specialized variant of Monte Carlo Tree Search with a transformer-based action encoder, a global depth-first search to break out of local minima, and repeated feedback from a model checker. We show that, under mild assumptions, this process will not miss correct solutions. In experimental validation, we show that GGMS can learn correct protocols for larger settings than existing methods.

## 1 Introduction

In modern distributed systems, we expect that many independently running computers can operate as if they were a single coherent system. This is fundamentally challenging because both individual computers and the networks that connect them are unreliable. *Distributed protocols*, i.e., sets of rules that govern communication and coordination between multiple processes in a distributed system, are a critical component in these systems. They provide guaranteed safety or correctness properties (e.g., that inconsistent states are never reached) that are needed by downstream tasks.

For example, a consensus protocol requires all correct processes to reach the same decision, despite that processes may start with different proposals and may fail during the protocol. There exist multiple variations of consensus protocols depending on their assumptions. A simple variation, the FloodSet algorithm (Lynch, 1996), works in multiple rounds: In each round, every process broadcasts its received proposals to others; at the end of the last round, every process decides to choose the received proposal with the minimal value (see Section 2.1 for details). Assuming there is a maximal number of $f$ failures, this protocol requires $f + 1$ rounds, ensuring that at least one round has no failure so that processes can reach the same state (i.e., received proposals).

Developing these protocols has often required decades of human effort. For example, complicated variations of consensus protocols, such as Paxos (Lamport, 1998; 2001; Moraru et al., 2013; Ongaro & Ousterhout, 2014) and Byzantine Fault Tolerance (Lamport et al., 1982; Castro & Liskov, 1999; Kotla et al., 2007; Giridharan et al., 2024), have motivated more than 40 years of research, which is still an active area. This work explores whether we can learn distributed protocols autonomously, without human knowledge beyond the desired properties of the target protocol. Our main contribution is algorithmic: we propose a synthesis framework that combines learning and model checking to construct distributed protocols with bounded formal guarantees.

We observe that designing a provably correct distributed protocol is similar to playing a board game: A process needs to choose an action so that, even if an opponent drives the system to the worst case, correctness is never violated. In a distributed protocol, this opponent could be a malicious process trying to break the protocol, or an "oracle" that can decide when processes fail or messages are lost. This raises the question of whether the methods developed that allow systems to master board

games with zero knowledge (Tesauro et al., 1995; Silver et al., 2017; Anthony et al., 2017; Brown & Sandholm, 2019b; Sudhakar et al., 2025) can be applied to learn distributed protocols.

To model distributed protocols in a generic way, we follow the wisdom in the distributed system community to **model a process in a distributed protocol as a state machine**. The state machine approach, which transits to the next state based on its input and current state, has been widely used (Schneider, 1990), which proves its generality. A human expert can specify the desired properties of a state machine, without knowing its internal details.

The major challenge we face is the combination of partial observability and the requirement of guaranteed correctness. Since processes can crash and messages can be lost, a process may not be able to observe the states of all other processes. However, we still expect that it can always make a correct decision. To address this challenge, we introduce Guided Global Monte Carlo Tree Search (GGMS), a new method design to learn correct distributed protocols with the following techniques:

**Validating the learned protocol.**   To achieve guaranteed correctness, we combine learning with model checking, which is widely used to verify the correctness of distributed protocols (Leesata-pornwongsa et al., 2014; Wang et al., 2023). If the check fails, we can use the counterexamples provided by model checking to further train the model.

**Ensuring convergence with global depth-first search.**   A standard AlphaGo-style procedure (that uses Monte Carlo Tree Search (MCTS) and a multi-layer perceptron to represent the policy) can occasionally converge to a correct protocol, but it is not guaranteed. We find that a key failure mode is that multiple correct protocols can be "**in superposition**". This occurs when different sets of initial states apply different correct protocols. The result is that some initial states at the border between protocols receive a mixture of protocols, which leads to validation failures. Because each policy has a set of supporting states where it is correct, further training does not break the impasse.

To address this, we propose a *global* depth-first search (DFS) procedure, where we systematically freeze the policy in a border state to a deterministic policy. If we then find that MCTS is unable to find a correct path for an initial state, we know that the current set of global freezes is inconsistent, and we undo the most recent freeze and continue the search in depth-first order.

By combining MCTS and DFS, we essentially rely on DFS to ensure convergence because DFS will eventually explore all the possible state machines, and rely on MCTS to provide hints about what transitions need to be frozen and to propagate the effects of such freezing to other transitions.

**Accelerating convergence with Guided MCTS.**   We propose multiple methods to accelerate convergence by guiding MCTS. First, since the internal states of a state machine have no specific meaning at the beginning, they can lead to many symmetric protocols. For example, swapping states A and B in one protocol can lead to another seemingly different protocol that is actually equivalent. This creates challenges for training convergence, which DFS will eventually resolve; however, it requires several DFS steps that increase with the number of protocol rounds and the number of states in each round. To accelerate this process, we freeze an entire MCTS path rather than a single decision at the beginning of DFS. We prove that this does not cause missing of correct protocols.

Second, in order for the combination of MCTS and DFS to work effectively, it is important to *propagate* the effects of the DFS freezes through MCTS. For example, a protocol may have a constraint that state A and state B should transit to the same state. Suppose DFS freezes state A to transit to state 1, we expect MCTS to learn that input B should also transit to state 1. However, due to the random and statistical nature of learning, such expected propagation does not always happen. Simulations that include both inputs A and B will find that B should transit to 1, but simulations of other scenarios that include only B may find that it is OK for B to transit to 0. If the former case happens rarely but the latter case happens a lot, training may lead to a high probability of B transiting to 0. Intuitively, we need sufficient "propagation power" (learning strength to generalize the logic of frozen transitions to related states) to overcome the "noise" created by random simulations..

To address this issue and build a strong propagation power, we follow two principles. First, after freezing, we only simulate scenarios where crashes happen in the last round and gradually relax this to allow crashes in earlier rounds. Second, after freezing, we only simulate the scenarios where the initial inputs lead to definite decisions, and then relax them to include ambiguous inputs that allow

different decisions. The basic idea behind these two principles is the same: When a protocol has inputs that lead to definite decisions, and when the protocol is in the last round, the correctness of its transitions is more certain than that in other scenarios. When training simulates only these scenarios, it can avoid random noise and build a strong propagation power. And after such a propagation is settled, we can then include other scenarios for further propagation.

We apply GGMS to learn an atomic commit protocol and a consensus protocol. Our evaluation shows that, compared with MCTS, GGMS has a much higher success rate (given a timeout) to learn a correct protocol. We further discuss future directions to address the limitations of our work.

## 2 BACKGROUND

### 2.1 DISTRIBUTED PROTOCOLS

A distributed protocol determines what messages a process should send and what actions a process should take when it receives certain messages. Depending on its goal, a distributed protocol is usually expected to achieve some correctness or safety properties (i.e., some constraints that its processes should never violate). These properties are defined by human experts. Depending on their assumptions, these protocols have different variations: Some variations assume that a failed process simply stops responding (i.e., omission failure), and some variations assume that a failed process can provide arbitrary responses (i.e., commission or Byzantine failure); some variations assume that network delay and clock drift are bounded (i.e., synchronous network), and some variations assume otherwise (i.e., asynchronous network).

To formally describe a distributed protocol, the distributed system community adopts the state machine approach, which models each process as a state machine. The state machine takes its current state and the incoming messages as input to determine the next state and what messages to send. With this approach, we can formally describe a distributed protocol and prove its properties.

**Example.** We use the FloodSet consensus algorithm (Lynch, 1996) as an example. As discussed in Section 1, a consensus algorithm assumes that different processes may start with different proposals but requires them to reach the same decision. Concretely, 1) a correct process should make a decision by the end of the protocol; 2) processes that made a decision should reach the same decision; 3) any decision must be an initial proposal from one process.

The FloodSet algorithm requires $f + 1$ ($f$ is the maximal number of processes that can fail) rounds of message exchanges. Each process $p$ maintains a set $W$, which is initialized as the initial proposal from $p$. In every round, each process broadcasts $W$ and adds all the proposals in the received sets to $W$. After $f + 1$ rounds, each process decides on the proposal with minimal (or any deterministic function) value from $W$. To describe this protocol using a state machine, the state machine has $W$ as its state. In every round, the state machine outputs $W$ to others and updates $W := W \cup \bigcup_j Received_j$. In the end, the state machine decides $min(W)$.

Consider an execution with three processes (P1-P3) and one failure (two rounds): Suppose P1, P2, and P3 start with proposals 1, 2, and 3 respectively. They broadcast such proposals to each other in Round 1, during which P1 fails. It may happen that P2 receives the message from P1 (and P3), so its $W$ becomes $\{1,2,3\}$ and P3 misses the message from P1 (but receives the message from P2), so its $W$ becomes $\{2,3\}$. In Round 2, P2 sends $\{1,2,3\}$ to P3, and P3 sends $\{2,3\}$ to P1, and since there is no failure, both end up with $\{1,2,3\}$ and decide 1.

This protocol can guarantee correctness due to the following reasoning: Among the $f + 1$ rounds, at least one round does not have process failures. In that round, all processes should reach the same $W$, which will not change in later rounds, and thus all processes should reach the same decision..

### 2.2 MODEL CHECKING AND FORMAL VERIFICATION

Given a state machine and a number of formally defined correctness properties, model checking techniques can validate that, for $N$ processes where $N$ is a concrete number, the correctness properties are never violated in any possible case. Brute-force model checking can achieve this by enumerating all possible cases. More efficient implementations leverage efficient SMT solvers like

Z3 (De Moura & Bjørner, 2008) and/or the specific properties of the target protocol (e.g., symmetry) (Leesatapornwongsa et al., 2014).

Using the consensus protocol as an example, for each process $n \in N$, and given its initial value $b_n$ and the set of final decisions $F$, we can formally define a number of desired correctness properties. We show one property as an example and document all properties in Section A.

**P1. No two processes should reach different decisions.**

$$\neg \exists n, m \ \in \ N \ \ \exists f_n, f_m \ \in \ F \ \left( f_n \ = \ \texttt{decision:0} \ \wedge \ f_m \ = \ \texttt{decision:1} \right) \quad (1)$$

Model checking can work with a concrete number of $N$ processes. As one can imagine, it takes longer to validate a protocol with a larger $N$. As a result, it becomes impractical after $N$ reaches a certain limit. On the other hand, formal verification techniques can prove that, for a generalized protocol with any number of $N$ processes, those properties will not be violated, usually through proof by induction (i.e., if a property holds for $n - 1$ processes, then prove that it also holds for $n$ processes (Hawblitzel et al., 2015; Ma et al., 2019; Yao et al., 2021; Zhang et al., 2025b)). Learning and proving the correctness of a generalized protocol is our future work.

## 3 Modeling a Distributed Protocol with Zero Knowledge

Following the widely used state machine approach, we model each participant process of a distributed protocol as a deterministic state machine. The input to a state machine includes its current state, messages received from other processes, and additional information like round number and process ID. The output of a state machine includes its new state and messages to be sent to other processes. A transition function determines the mapping between an input and an output (see examples in Section 2.1). After a state machine applies a transition function, it will update its state to be the new state in the output.

We categorize the state of a state machine into four types: initial states, decision states, lost states, and internal states. Initial states are the states a process may start with. Decision states are the states a process may expose to the external world. These two types of states must be defined by the human expert, because they are related to how to define the correctness properties of the protocol. Lost state is a special state indicating a message is lost. The state machine may receive the lost state in its input, but should never output the lost state. Internal states are the states a process may use internally for bookkeeping. A protocol is free to use any number of internal states and specify their meanings. For example, in the primary-backup protocol discussed in Section 2.1, initial states include `init:0` and `init:1`; decision states include `decision:0` and `decision:1`; internal states include `proposal:0` and `proposal:1`.

Modeling a participant as a state machine enables the zero-knowledge learning approach introduced by AlphaGo-Zero, which does not require human data or knowledge beyond "game rules": A human expert only needs to define the "game rules", i.e., initial states, decision states, and correctness properties on these states, but does not need to make any suggestions about how the state machine may work internally, i.e., the number of internal states and the transitions between any states. It also does not require any execution traces from a correct protocol.

Given a state machine to model one process, we make several assumptions about how the whole distributed protocol works. We assume that every process follows the same state machine. We assume that the protocol works in multiple rounds. Before the first round, the state machine of each process will be given an initial state, set up by our training procedure. At the beginning of each round, a process will broadcast its own state to every process, including itself, while some of the messages may be lost due to process crashes. Then each process will apply its state machine, using the received states as the input, to determine the next state of the state machine. A process will broadcast the updated state in the next round. We also assume that a process can only go to the decision state in the last round.

Figure 1 shows an example execution following the idea of the FloodSet protocol, given a state machine, initial states, and some crashes. We give every state a unique value as shown in the figure. In this example, all processes start with the same initial state `init:1(6)`. In the first round, every process broadcasts its initial state to all others, while Process 3 fails during this round. As a result,

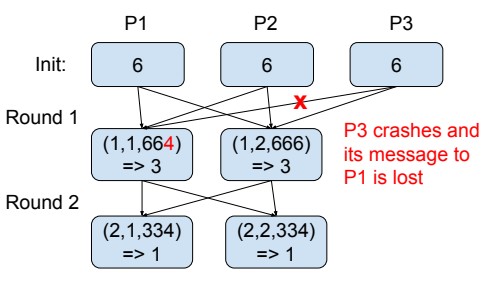

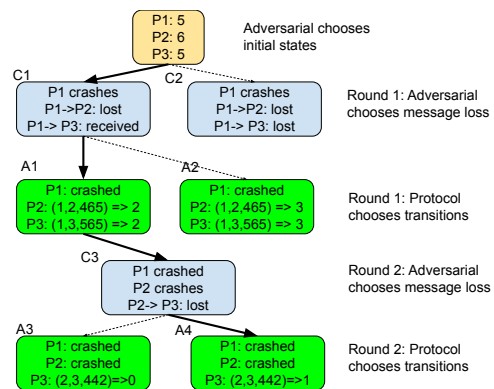

Figure 1: Simulating the FloodSet protocol. P3 crashes in Round 1 and causes P1 and P2 to have diverged inputs, but P1 and P2 still converge eventually following the protocol.

Figure 2: Applying MCTS. Adversarial chooses initial states and lost messages that are likely to violate correctness. Protocol chooses transitions that are likely to achieve correctness.

Process 1 does not receive the message `init:1(6)` from Process 3 and thus it constructs the input `(1,1,664)` (first 6 from itself, second 6 from Process 2, and the 4 denotes a lost message from Process 3) as its input to its state machine. Process 2, on the other hand, receives the message 6 from Process 3, so its input state is `(1,2,666)`. Each process then independently applies its state machine. In this example, both `(1,1,664)` and `(1,2,666)` transition to the same state, `internal:a(3)`. In the second round, the remaining alive processes exchange their current state again and apply the state machine to determine the final decision (i.e.,`decision:1(1)` for both).

These assumptions introduce a few limitations that we plan to address in the future. First, the assumption that every process follows the same state machine makes it impossible to model protocols that may include Byzantine processes. Second, the assumption that a process will broadcast its own state to every process in every round makes it impossible to model a number of categories of protocols, e.g., a process sends different messages to different processes. Third, we make the synchronous networking assumption that if a message is sent successfully at the beginning of round $i$, it will be received successfully at the end of round $i$. Of course, this assumption won't hold for asynchronous protocols like Paxos. Finally, the assumption that a process can only make a decision in the last round makes it impossible to model protocols that can make early decisions or multiple decisions. Nevertheless, despite these limitations, we can model a number of important protocols. We discuss possible relaxation of these assumptions in Section 6.

With the above intuitions, we formally define our problem. A target distributed protocol is defined as a tuple $DP = \{I, D, C\}$, with the set of input states $I$, the set of decision states $D$, and the set of desired correctness properties $C$. A **setting** is defined as a tuple $S = \{n, r, f, in\}$, with the number of processes $n$, the number of rounds $r$, the number of processes allowed to fail $f$, and the number of internal states $in$. A **scenario** is defined as $Sc = \{Init, Loss\}$, where $Init$ is a vector of $n$ values, denoting the initial state of each process, and $Loss$ is a set of tuples $\{id1, id2, i\}$, denoting the message from process $id1$ to process $id2$ is lost during round $i$. Note that we do not include crashes into the scenario, because a crash of a process will be observed as message losses at other processes. A **state machine** $SM$ is defined to include a $state$ and a set of transition rules $[round, procID, inputStates] \rightarrow newState$, where $round$ denotes the round number, $procID$ denotes the process ID, and $inputStates$ is a vector denoting the input from all processes.

Given $DP$, $S$, $Sc$, and $SM$, a **simulation** follows the following steps: 1) At the beginning, it sets $SM[id].state$ to $Init[id]$. 2) In every round i, it computes the $inputStates$ of each $SM$ that did not send a loss message in prior rounds (i.e., not crashed): If $Loss$ does not contain $\{id1, id2, i\}$, then $SM[id2].inputStates[id1] = SM[id1].state$; otherwise, $SM[id2].inputStates[id1] = L$ ($L$ is the lost state). 3) It then applies the transition for each process: $SM[id].state = transition(i, id, SM[id].inputStates)$. Given $DP$ and $S$, our goal is to find an $SM$, such that for any $Sc$, simulation($DP$,$S$,$Sc$,$SM$) does not violate $C$.

We say a setting $S$ is **feasible** if there exists a correct state machine for that setting. Without prior knowledge, our training does not know which settings are feasible. Our strategy is to start from smaller settings and add more states and/or rounds if we cannot learn a correct state machine.

## 4 LEARNING A DISTRIBUTED PROTOCOL

### 4.1 MONTE-CARLO TREE SEARCH

Inspired by the similarity between distributed protocols and board games, we first apply the Monte Carlo Tree Search (MCTS) simulation approach.

We model the whole process as two players in a game. The protocol player tries to find the right transitions in its state machine so that all processes in the distributed protocol can always achieve the correctness properties. The adversarial player tries to find the scenarios that can defeat the protocol player's state machine, i.e. making it violate the correctness properties. While it may be possible to train a model for the adversarial player as well, our current implementation relies on random exploration for the adversarial player and relies on the model checker to make the final checking.

The whole simulation process works as shown in Figure 2. In one simulation, the adversarial player first selects the initial state for each process and messages to lose for the first round; our simulation follows the procedure in Section 3 to let each process broadcast its state, apply the message loss selected by the adversarial player, and compute the input for each process; then the protocol player selects the new state for each process based on its input. The simulation repeats this process until the maximum number of rounds is reached. By executing the simulation multiple times, allowing both players to explore different transitions, we can merge their results to build a search tree (Figure 2).

Our MCTS implementation is similar to that of AlphaGo-Zero, with the following differences. First, we do not use the value network in our MCTS. In AlphaGo-Zero, the value network is used to predict the expected reward, which guides the MCTS simulations. However, the number of rounds of the distributed protocol is much smaller than that in Go. This means we can simulate until the end of a protocol to get the real reward without relying on predictions from the value network.

Second, in AlphaGo-Zero, the MCTS algorithm is also used for inference, i.e., when actually playing the game or running the protocol. However, we can only use the trained policy network for inference because the internal information, such as visited count, rewards, and probabilities, is invisible between different processes during actual running. In other words, during inference, among multiple transitions from the same input, a process will always apply the one with the highest probability.

The details of our MCTS implementation can be found in Section C.5.

### 4.2 ENSURING CONVERGENCE WITH GLOBAL DEPTH FIRST SEARCH

The MCTS approach alone can occasionally converge to a correct state machine, but often fails to do so. Our investigation shows that the primary reason is the superposition problem, that is, there often exist multiple versions of correct state machines, and MCTS may end up in a situation where it learns some transitions from one version and some other transitions from another version, but when these transitions are combined, they do not generate a correct state machine.

There are multiple reasons for the existence of multiple correct versions. First, the state machine has the flexibility to give a specific meaning to an arbitrary internal state, creating multiple equivalent protocols. For example, for the consensus problem, one version could use internal state `internal:a` to represent the intention for `decision:0`, and another version could use `internal:b` for the same purpose.

Second, some protocols inherently have some flexibility to allow different decisions. For example, for consensus, if some processes have `init:0` and some have `init:1`, they could either all decide `decision:0` or all decide `decision:1`. Suppose that in such a scenario, a process has input A and another process has input B at the same round. In a correct protocol, both A and B should transit to the same state, either `intent-0` or `intent-1`. When simulating this scenario, MCTS can find this constraint and maybe give $A \rightarrow$ `intent-0` and $B \rightarrow$ `intent-0` a higher probability than $A \rightarrow$ `intent-1` and $B \rightarrow$ `intent-1`. However, A or B can also appear alone in other scenarios.

When combing probabilities across scenarios, MCTS may end up with $A \to$ intent-0 having a higher probability than $A \to$ intent-1 but $B \to$ intent-1 having a higher probability than $B \to$ intent-0, which violates the constraint that A and B should transit to the same state. We show detailed examples of these two reasons and how they affect MCTS in Section D.

Our approach, Guided Global Monte Carlo Tree Search (GGMS), addresses this issue by enhancing MCTS with Depth-First Search (DFS). The key idea is that, if multiple transitions from the same input have similar probability, GGMS should try to freeze it to one transition (i.e., don't allow random exploration for this input during MCTS) and then keep training to see whether it can get a correct state machine. In the above example, by freezing the transition $A \to$ intent-0, we hope that keeping training will motivate $B \to$ intent-0. Since there might be multiple such ambiguity points, GGMS may repeat freezing multiple times. And since freezing may be wrong (e.g., freezing $A \to$ intent-0 and $B \to$ intent-1 for whatever reason), GGMS also needs to unfreeze certain transitions when it hits a dead end. This procedure leads to a DFS-like search, in which GGMS keeps freezing certain transitions until either it succeeds in finding a correct state machine or it hits a dead end. In the latter case, it unfreezes prior frozen transitions in an DFS manner.

**Theorem 1** *With a feasible setting, assuming that GGMS's unfreezing condition is accurate (i.e., it does not unfreeze when prior frozen transitions are part of a correct state machine), GGMS can eventually find a correct state machine.*

This is because, for a specific setting, the number of possible state machines is finite. This means that DFS can eventually explore all possible state machines, ensuring that we can find a correct one.

However, naive DFS (i.e., randomly freezing a transition) may take too long. Consider a protocol which involves two initial states, two internal states, two decision states, three processes, and three rounds, and assume that process ID does not affect transitions, the total number of possible inputs to the state machine is $3(round) \times 2(ownState) \times 3^2(otherState) = 54$. And since each of these inputs can transit to two values, the total number of state machines is $2^{54}$. Using DFS to explore each state machine is too expensive. By combining DFS and MCTS, GGMS relies on DFS to break ambiguity, and relies on MCTS to provide hints about what transitions to freeze and "propagate" the effect of freezing (e.g., if we freeze $A \to$ intent-0, then MCTS can find that $B \to$ intent-0).

We present the details of our DFS algorithm in Section C.3, which includes the conditions for freezing and unfreezing and how to determine which transitions to freeze or unfreeze. Our current unfreezing condition, which is based on exhaustive search, is accurate but not scalable, and we discuss possible ways to replace exhaustive search with a more scalable Z3-based solver.

Note that in practice, there is always a time limit for training, so despite the eventual convergence guarantee, GGMS may not succeed within the time limit. However, this property means that we can always devote more time and/or resources to increase the chance of success.

### 4.3 ACCELERATING CONVERGENCE WITH GUIDED MCTS

While DFS helps to address the ambiguity issue, its speed is sometimes not satisfactory due to the following reasons. First, as discussed before, since GGMS can give a meaning to an arbitrary internal state, it needs multiple rounds of freezing to break the ambiguity among them. With more rounds in the protocol and more states, this process requires more rounds of freezing. To address this problem, at the beginning of training, GGMS freezes all transitions in one particular scenario (S), so that the simulation can reach a correct decision. This approach helps GGMS to break the ambiguity among internal states more quickly. Furthermore, we can prove that, under certain conditions, it will not hurt the capability of GGMS to find a correct state machine.

**Theorem 2** *In a feasible setting, assuming that 1) the scenario of this particular simulation leads to a definite decision (i.e, no ambiguity), and 2) for each pair of (round, procID), this approach only freezes one transition (round, procID, inputA) $\to$ B, then there exists a correct state machine with all the frozen transitions ($Transitions_{fix}$).*

We provide the formal proof in Section B. Intuitively, if a correct state machine has (round, procID, inputA) $\to$ C, we can always swap B and C for (round, procID) to get an equivalent protocol.

Second, as discussed before, we expect MCTS to propagate the effect of freezing. In the above example, when GGMS freezes $A \rightarrow$ `intent-0`, it expects MCTS to find that it should choose $B \rightarrow$ `intent-0`. In practice, such propagation does not always succeed, since propagation often relies on a particular scenario to identify the relationship. In the above example, in order for GGMS to learn that A and B should transit to the same state, A and B should happen together in the same simulation. In other scenarios where only B occurs, GGMS may find that it is OK for $B$ to transit to either `intent-1` or `intent-0`, due to the ambiguity problem discussed above. If simulations including both A and B happen rarely, but simulations including only B happen often, GGMS may not give $B \rightarrow$ `intent-0` a high probability. This problem is particularly troublesome when the random initialization of the state machine gives $B \rightarrow$ `intent-1` a high probability to begin with.

To address this issue, we introduce a guided sampling method. After freezing the initial path, GGMS simulates only those scenarios where message losses occur in the final round and only the initial states that lead to a definite decision. After the model becomes fully correct in this stage, GGMS relaxes the restriction on the message loss pattern to allow losses in the last two rounds. GGMS then repeats the simulation and learning process until the model achieves full correctness under this relaxed setting. GGMS repeats this until it allows message losses in all rounds. Eventually, GGMS relaxes the simulation to consider all possible initial states.

Such a design is based on the observation that there is less ambiguity when the protocol is in later rounds and when the protocol starts with initial states that lead to definite decisions. By first simulating in these scenarios, GGMS can better avoid the "noise" from ambiguity, and thus better propagate the effects of freezing. Then, after such propagating has settled (i.e., the corresponding transitions reach a high probability), GGMS can further propagate its effects by relaxing crash and initial states scenarios. We present the details in Section C.1.

### 4.4 Validating the Learned Model

In our current implementation, we use the brute-force verifier that enumerates all the possible scenarios. We find that it is not the bottleneck, as MCTS takes most of the time. In the future, assuming MCTS will be optimized, we will switch to more efficient verifiers like Z3.

After each training episode, GGMS updates the model based on MCTS results. Then GGMS applies the verifier to validate the model: If validation does not find any counterexamples, GGMS either terminates or relaxes to explore more scenarios. Otherwise, GGMS outputs all counterexamples. In the next episode, GGMS increases the sampling rate of the scenarios of these counterexamples. Section C.4 presents the details of our brute-force verifier and Section C.2 presents how GGMS applies counterexamples in the following training.

## 5 Evaluation

We apply GGMS to learn synchronous atomic commit and consensus protocols. Atomic commit may look similar to consensus to some extent: Processes start with proposals "abort" or "commit" and try to reach the same decision. However, atomic commit is different from consensus in two ways: First, if a process proposes "abort", then the final decision of atomic commit must be "abort" (consensus allows "commit" if another process proposes "commit"). Second, if a process crashes, the final decision could be "abort", even if all processes propose "commit" (consensus requires "commit" in this case). Such differences lead to protocols that are different from consensus.

We document the formal properties of these protocols in Section A and our experiment settings in Sections E and F. Synchronous consensus is a well-studied field, with protocols such as Flood-Set and Primary Backup (Bressoud & Schneider, 1996). Synchronous atomic commit is not well-studied as far as we know, because well-known atomic commit protocols, such as two-phase commit (2PC) (Gray, 1978), target asynchronous environment.

For synchronous consensus, human knowledge tells that a setting $S$ with $in = 2$ ($in$ is the number of internal states) and $r > f$ ($r$ is the number of rounds and $f$ is the number of failures allowed) is feasible. We verified the same conclusion for synchronous atomic commit. We report results on such feasible settings. As a sanity check, we tested some infeasible settings and did not get a correct protocol. In the following report, we use a triple prot-$n$-$f$ to represent the setting: prot represents

| | Success rate | | | Running time (minutes) | | | | | | | | |
|---|---|---|---|---|---|---|---|---|---|---|---|---|
| | MCTS | MCTS+DFS | GGMS | MCTS | | | MCTS+DFS | | | GGMS | | |
| | | | | avg | min | max | avg | min | max | avg | min | max |
| ac-2-1 | 50% | 60% | 100% | 133 | 15 | 301 | 52 | 25 | 96 | 15 | 8 | 29 |
| ac-3-2 | 0 | 40% | 60% | – | – | – | 1413 | 722 | 1825 | 813 | 175 | 1441 |
| ac-4-1 | 10% | 20% | 100% | 184 | 184 | 184 | 192 | 191 | 193 | 413 | 310 | 630 |
| ac-4-2 | 0 | 0 | 70% | – | – | – | – | – | – | 754 | 563 | 938 |
| con-2-1 | 90% | 90% | 100% | 46 | 3 | 180 | 25 | 3 | 55 | 8 | 7 | 11 |
| con-3-2 | 80% | 80% | 100% | 1106 | 49 | 2387 | 273 | 148 | 693 | 118 | 104 | 151 |
| con-4-1 | 80% | 70% | 100% | 328 | 76 | 1576 | 348 | 146 | 654 | 268 | 253 | 280 |
| con-4-2 | 30% | 40% | 90% | 678 | 470 | 922 | 1783 | 1277 | 2560 | 717 | 610 | 970 |
| con-4-3 | 0 | 0 | 100% | – | – | – | – | – | – | 3358 | 1857 | 5693 |

Table 1: Success rate and running time of different methods. We run each setting 10 times.

the protocol ("ac" for atomic commit and "con" for consensus). $n$ is the total number of processes. We set $r = f + 1$. We compare GGMS with pure MCTS and MCTS+DFS to understand the effectiveness of DFS and Guided MCTS. MCTS includes the techniques described in Section 4.1; MCTS+DFS include Sections 4.1 and 4.2; GGMS includes Sections 4.1, 4.2, and 4.3.

**Success rate.**    Table 1 shows the success rate of different methods. As DFS can guarantee eventual success, this success rate is based on a limited time, which is documented in Sections F. For one setting, we run a method 10 times and report the number of times it can succeed in finding a correct protocol. For MCTS and MCTS+DFS, if they had a low success rate at a certain setting, we did not further try them on larger settings. As shown in this figure, in all settings, GGMS consistently achieves higher success rates than MCTS and MCTS+DFS. From the training logs, we find that MCTS is primarily bothered by the superposition problem, and increasing training time does not help much. MCTS+DFS tries to address the superposition problem by freezing transitions, but when it freezes wrong ones, it will need unfreezing, which takes a lot of time.

**Running time.**    Table 1 shows the running time to converge to a correct model (excluding unsuccessful runs). GGMS' speed is faster than or similar to that of MCTS and MCTS+DFS in most of the settings. Note that since we exclude unsuccessful runs, this is favorable for MCTS and MCTS+DFS: In a difficult setting, MCTS and MCTS+DFS may fail many trials, which do not count, but GGMS may succeed after a long run, which increases its average running time.

Our further analysis shows that in GGMS, the simulation time for MCTS dominates the overall time compared to the training and validation time. For atomic commit, we also observe a significant variation in running time, usually due to incorrect freezing leading to unfreezing.

**Scalability.**    In general, the running time increases rapidly with larger settings, which is expected and makes it hard to learn large settings. We currently implement GGMS with a single threaded Python program (except the verifier, which is parallelized). We will apply techniques including implementing performance-critical steps in C, parallelizing performance-critical steps, and maybe further accelerating it with GPUs. We expect such optimizations to bring significant performance improvement. However, we will still see the exponential growth of the running time with an increasing number of processes. As a result, our speculation is that we may be able to scale to 8-10 processes for some protocols, but probably not to 100 processes.

However, we do not think scaling to a large number of processes is necessary to design a distributed protocol. For human experts, a general practice is to 1) first design a protocol for a small number of processes, then 2) distill the insights from these small instances and generalize these insights for a generic protocol with an arbitrary number of processes, and finally 3) prove the correctness of the generic protocol. This process assumes that the key idea of a generic protocol can be observed at small scales. Therefore, as long as our tool can accomplish step 1), it may provide useful insights for human experts. Automating step 2) is our future work. Prior work has done such automated extrapolation in related but different fields (e.g., Ma et al. (2019); Yao et al. (2021); Zhang et al. (2025b)). Automating 3) is quite well established in the distributed system community.

**Found protocols.** For consensus, GGMS finds protocols that are similar to the FloodSet protocol. For atomic commit, GGMS finds protocols that modify FloodSet to adapt to the additional requirement of atomic commit. Concretely, these protocols treat "lost message" as "abort" in the first round and ignore "lost message" in later rounds. Though simple, this is a new protocol as far as we know.

We present more results in Section G, including how the number of counterexamples changes with training and how ML models affect our results.

**Comparison with GPT5.1** We used GPT5.1 with extended thinking as a synthesizer, manually providing counterexamples with each attempt. While GPT finds the FloodSet protocol for conensus, it failed to synthesize a compliant state machine for atomic commit (Appendix I).

## 6 FUTURE WORK

Section 3 discusses several limitations of this work. Allowing a process to make a decision early before the last round is a straightforward extension of this work. Allowing a process to send different messages to different processes will significantly increase the search space. As a middle ground, we may allow a process to send its state to a subset of processes. This approach will not increase the search space too much, but will help us model more practical protocols. To model asynchronous networking, we need to change our simulator to include more complicated message loss scenarios. To model Byzantine processes, we can incorporate adversarial models so that two models play against each other. In the long term, it is also interesting to explore whether we can derive a general protocol that can work with any number of processes from the protocols we learned.

## 7 RELATED WORK

Modern ML systems that combine self-play with search have achieved superhuman performance in games with both full and partial observability (Mnih et al., 2015; Silver et al., 2017; Li et al., 2020; Bard et al., 2020; Brown & Sandholm, 2019a). These methods typically optimize expected reward and output neural policies without formal guarantees. Designing distributed protocols is harder in two ways: processes have only local views (partial observability), and we require *guaranteed correctness* under faults and message loss rather than high average-case performance. To the best of our knowledge, prior work does not learn distributed protocols with formal correctness guarantees.

Closer to our problem, Khanchandani et al. (2021) use self-play to learn algorithms for a distributed directory problem, while Zhang et al. (2025a) and Belardinelli et al. (2024) combine reinforcement learning with model checking for multi-agent systems. These approaches learn policies or algorithms and use verification as a shaping signal or evaluation tool. In contrast, GGMS searches directly in a symbolic space of global state machines for synchronous, crash-prone message-passing protocols, and uses model checking as a hard oracle that accepts or rejects entire candidate protocols.

Classical distributed protocols and verification frameworks (Gray, 1978; Corbett et al., 2012; Lamport, 2001; Ongaro & Ousterhout, 2014; Castro & Liskov, 1999; Hawblitzel et al., 2015; Ma et al., 2019; Yao et al., 2021; Zhang et al., 2025b), automata learning (de la Higuera, 2005; Heule & Verwer, 2010), and counterexample-guided inductive synthesis (CEGIS) (Solar-Lezama, 2008) provide important context: they either assume a human-designed protocol and prove its correctness, or synthesize sequential programs or automata from examples and counterexamples. GGMS follows a CEGIS-like loop but applies it to distributed protocol automata, guided by RL and MCTS under an asynchronous fault model. Appendix H further situates GGMS within this landscape, including comparisons to recent large-scale reasoning and coding agents such as AlphaProof and AlphaEvolve (Hubert et al., 2025; Novikov et al., 2025).

## 8 CONCLUSION

This work explores the new area of learning provably correct distributed protocols with partial observability. We propose GGMS, which combines model checking to ensure correctness, DFS to ensure eventual convergence, and guided MCTS to accelerate convergence. As a preliminary exploration, we further discuss possible future directions.

## 9 REPRODUCIBILITY

We submit our source code as a zip file in the supplementary materials. Our work does not rely on any data set.

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

# A    FORMAL DEFINITIONS OF INVESTIGATED PROTOCOLS

The complete definitions of each state are provided in Table S1 and Table S2, for the atomic commit and consensus protocols, respectively. Based on these state definitions, we formally define the properties of each protocol following prior work. Let $N$ denote the set of all processes. Each process has an initial state $b_n$, and a set of final decisions, $F_n$. Let $M$ represent all messages exchanged between processes, and let $L = \exists m \in M(m = Lost)$ indicate that there exists at least one lost message.

| State | Meaning |
|---|---|
| `init:abort` | Initial state representing intent to abort |
| `init:commit` | Initial state representing intent to commit |
| `internal:a` | Internal state |
| `internal:b` | Internal state |
| `decision:abort` | Final decision to abort |
| `decision:commit` | Final decision to commit |

Table S1: State Definitions for Atomic Commit Protocol

| State | Meaning |
|---|---|
| `init:0` | Initial state representing intent to commit 0 |
| `init:1` | Initial state representing intent to commit 1 |
| `internal:a` | Internal state |
| `internal:b` | Internal state |
| `decision:0` | Final decision to commit 0 |
| `decision:1` | Final decision to commit 1 |

Table S2: State Definitions for Consensus Protocol

**Atomic commit protocol properties.**    Rule P1 states that no two processes should reach conflicting decisions. Rule P2 ensures that if all processes initially intend to commit and there are no message losses, then all processes must reach a commit decision. Rule P3 indicates that if any process initially intends to abort, then all processes must reach an abort decision. Rule P4 requires that every process must eventually reach a final decision.

**P1.**

$$\neg \exists n, m \in N \ \exists f_n, f_m \in F$$
$$\left( f_n = \texttt{decision:abort} \wedge f_m = \texttt{decision:commit} \right) \quad (2)$$

**P2.**

$$\left( \forall n \in N, \ b_n = \texttt{init:commit} \right) \wedge (\neg L)$$
$$\Rightarrow \left( \forall n \in N, \ \forall f_n \in F_n, \ f_n = \texttt{decision:commit} \right). \quad (3)$$

**P3.**

$$\left( \exists n \in N, \ b_n = \texttt{init:abort} \right)$$
$$\Rightarrow \left( \forall n \in N, \ \forall f_n \in F_n, \ f_n = \texttt{decision:abort} \right) \quad (4)$$

**P4.**

$$\forall n \in N, \exists f_n \in F \ \left( f_n = \texttt{decision:abort} \vee f_n = \texttt{decision:commit} \right) \quad (5)$$

Note that the well-known atomic protocols like two-phase commit (2PC) and three-phase commit (3PC) make the asynchronous networking assumptions that every message has a non-zero chance of being lost. As discussed in Section 3, our current implementation supports only synchronous networking assumptions. Such a difference leads to a difference in the protocol properties: In the asynchronous versions, we can prove that there always exist scenarios that a correct process cannot decide. In the synchronous versions, such scenarios do not exist.

**Consensus protocol properties.** Rule P1 states that no two processes should reach conflicting decisions. Rules P2 and P3 assert that if all processes start with the same initial proposal, then all final decisions must match that proposed value. Rule P4 requires that every process must reach a decision by the end of the protocol.

**P1.**

$$\neg \exists n, m \in N \; \exists f_n, f_m \in F \; \left( f_n = \texttt{decision:0} \land f_m = \texttt{decision:1} \right) \quad (6)$$

**P2.**

$$(\forall n \in N, b_n = \texttt{init:1}) \qquad \Rightarrow \qquad (\forall f_n \in F, f_n = \texttt{decision:1}) \quad (7)$$

**P3.**

$$(\forall n \in N, b_n = \texttt{init:0}) \qquad \Rightarrow \qquad (\forall f_n \in F, f_n = \texttt{decision:0}) \quad (8)$$

**P4.**

$$\forall n \in N, \exists f_n \in F \; \left( f_n = \texttt{decision:0} \lor f_n = \texttt{decision:1} \right) \quad (9)$$

## B  PROOF OF THEOREM

**Theorem 2** *In a feasible setting, assuming 1) the scenario Sc of this particular simulation leads to a definite decision (i.e, no ambiguity), and 2) for each pair of (round, procID), this step only fixes one transition (round, procID, inputA) → B, then there exists a correct state machine with all the fixed transitions ($Transitions_{fix}$) in this step.*

Assuming there exists a correct state machine $SM_0$, we prove that we can construct a state machine $SM_{r-1}$, such that, 1) $SM_0$ and $SM_{r-1}$ are equivalent, i.e., for every scenario $S$, every non-crashed process makes the same decision with $SM_0$ and $SM_{r-1}$; and 2) $SM_{r-1}$ includes all the transitions in $Transitions_{fix}$.

We construct by induction of $r - 1$ steps ($r$ is the number of maximal rounds). Assuming $SM_{i-1}$ is already constructed ($1 < i < r$), we construct $SM_i$ in the following way. We search for procID, such that transition $[round = i, procID, inputA] \rightarrow B$ exists in $Transitions_{fix}$ and $[round = i, procID, inputA] \rightarrow C$ exists in $SM_{i-1}$ and B is different from C. Then we swap B and C in $SM_{i-1}$ to get $SM_i$: 1) For any input, if there exists a transition $[i, procID, input] \rightarrow C/B$ in $SM_{i-1}$, we change it to $[i, procID, input] \rightarrow B/C$ in $SM_i$; 2) In round i+1, for any procID2, if there exists a transition $[i + 1, procID2, input] \rightarrow D$ in $SM_i$, where the input vector includes C (or B) from procID, we change B into C and C into B in the input vector.

First, we can prove that $SM_i$ includes transitions in $Transitions_{fix}$ whose round is smaller than i+1. We prove by induction. When $i = 1$, $SM_1$ is constructed from $SM_0$. Under $Sc$, $SM_0$ and $Transitions_{fix}$ must reach the same $[round = 1, procID, inputA]$ in the first round, since the input is from initial states. And if a $[round = 1, procID, inputA]$ transit to different states in $SM_1$ and $Transitions_{fix}$, our construction changes the output of the transition in $SM_1$ to match that in $Transitions_{fix}$. Then for the later round i, we can prove it in the same way. Under $Sc$, $Transitions_{fix}$ and $SM_{i-1}$ must reach the same $[round = i, procID, inputA]$ in round $i - 1$, as they have the same transitions for $Sc$. Then if $[round = i, procID, inputA]$ transit to different states in $Transitions_{fix}$ and $SM_{i-1}$, our construction forces them to be the same in $Transitions_{fix}$ and $SM_i$ Note that this only works if for each pair of (round, procID), $Transitions_{fix}$ only includes one transition. Otherwise, the construction may need to swap multiple pairs of values, and they may conflict, which means the construction may not be possible (e.g., we cannot both swap B and C and swap B and D).

Second, we can prove that $SM_{i-1}$ and $SM_i$ are equivalent. Assuming a simulation applies transition $(i, procID, input) \rightarrow B/C$ in $SM_{i-1}$, the simulation will get the same input for procID in round i when applying $SM_i$, as $SM_{i-1}$ and $SM_i$ have the same set of transitions before round i. Then the simulation will apply $(i, procID, input) \rightarrow C/B$ in $SM_i$, i.e., output of $SM_{i-1}$ and $SM_i$ swap B and C for procID. However, since our construction also swaps B and C in the input of round i+1 from

---

**Algorithm 1:** Pseudo code of GGMS

1   $model \leftarrow$ init_model();
2   $phase\_ID \leftarrow 0$;
3   $training\_buffer \leftarrow []$;
4   $failed\_scenarios \leftarrow []$;
5   $freeze\_list \leftarrow []$;
    /* Each main loop iteration is an episode                       */
6   **while** *true* **do**
7     **for** $i \leftarrow 1$ **to** $100$ **do**
8         $scenario \leftarrow$ sample_scenario($phase\_ID, failed\_scenarios$);
9         $training\_data, reward \leftarrow$ run_mcts($scenario, model$);
10         $training\_buffer$.append($training\_data$);
11         determine_unfreeze($reward, freeze\_list$);
12     **end**
13     determine_freezing($training\_buffer, freeze\_list$);
14     update_model($training\_buffer, model$);
15     $failed\_scenarios \leftarrow$ validate($phase\_ID, model$);
16     **if** $failed\_scenarios == [\,]$ **then**
17         **if** $phase\_ID == lastPhase$ **then**
18           terminate;
19         **end**
20         **else**
21           $phase\_ID \leftarrow phase\_ID + 1$;
22         **end**
23     **end**
24   **end**

---

procID, we can know that in round i+1, every process will still transit to the same state. Therefore, $SM_{i-1}$ and $SM_i$ are equivalent.

With the above steps, we can construct a state machine $SM_{r-1}$ that is equivalent to $SM_0$ and that includes all the transitions from $Transitions_{fix}$ till round r-1. In round r, every state machine transits to the decision state. We can prove that $SM_{r-1}$ must have the same transitions as those in $Transitions_{fix}$ for round r, with no need for swapping. This is because, for $Sc$ that generates $Transitions_{fix}$, $SM_{r-1}$, and $Transitions_{fix}$ will apply the same transitions till round r-1, so every process should have the same input at the beginning of round r. If $SM_{r-1}$ and $Transitions_{fix}$ have different transitions for the same input in round r, they will lead to different decisions of some processes. This contradicts our assumption that this scenario leads to a definite decision.

Therefore, we have proved that $SM_{r-1}$, which is equivalent to $SM_0$ and must be correct, has all the transitions of $Transitions_{fix}$.

## C   IMPLEMENTATION DETAILS

### C.1   OVERVIEW

Algorithm 1 presents the high-level pseudo code of GGMS.

model is a neural network representing the state machine we want to learn. As discussed in Section 3, it takes $[round, procID, inputStates]$ as the input and outputs a $newState$. In fact, to facilitate learning, we let it output a probability for each potential value of $newState$. During inference, GGMS will choose the value with the highest probability.

phase_ID is used to implement the guided MCTS (Section 4.3). When set to 0, it means that GGMS will only simulate scenarios with definite initial states and message losses in the last round. When set to 1, it means that GGMS will simulate scenarios with definite initial states and message losses in the last two rounds, etc. Finally, when set to $r$ (the total number of rounds), it means that GGMS can use any initial state and message losses in any round.

---

**Algorithm 2:** Pseudo code of choosing a scenario

```
1  Function sample_scenario(phase_ID, failed_scenarios):
2      u ← Uniform(0, 1);
3      if u < 0.3 or failed_scenarios == [] then
4          scenario ← uniform_sample(generate_all_scenarios(phase_ID));
5      end
6      else
7          scenario ← uniform_sample(failed_scenarios);
8      end
9      return scenario;
10 end
```

---

`training_buffer` is a bounded buffer to store training data collected during MCTS. Each item in the buffer records the probability of a transition from $[round, procID, inputStates]$ to one potential output value.

`failed_scenarios` records scenarios that caused prior validation to fail. As discussed, GGMS will use such scenarios to retrain the model.

`freeze_list` is used to implement DFS (Section 4.2). Like a conventional DFS implementation, `freeze_list` is a stack, and each item in the stack represents one frozen transition $[round, procID, inputStates] \rightarrow newState$. Each item also records whether other values have been frozen for the same $[round, procID, inputStates]$ in the past.

The whole algorithm works in multiple episodes. In each episode, it runs MCTS on 100 scenarios (line 7). Each scenario is either randomly chosen from scenarios allowed by the current phase or from past failed scenarios (line 8). Running MCTS on the scenario will generate some training data, which will be added to the training buffer, and a reward (lines 9-10). GGMS will determine whether it needs to unfreeze depending on the reward (line 11).

Then, after simulating 100 scenarios, GGMS will determine whether it needs to freeze more transitions based on the new training data (line 13). Note that unfreeze and freeze are determined at different timings: If MCTS cannot find a good model for one scenario, that is already enough to trigger unfreeze, and that is why unfreeze is determined after simulating every scenario. However, determining freezing often requires information from multiple scenarios, which can lead to potentially different transitions for the same input, and that is why GGMS determines freezing after trying a number of scenarios.

Then GGMS updates the model using the new training data (line 13) and then validates the new model (line 14). If validation does not find any failed scenarios, GGMS will either terminate if this is the last phase, or proceed to the next phase otherwise (lines 16-23).

## C.2 USE COUNTEREXAMPLES TO RETRAIN

Algorithm 2 presents the details of how GGMS selects scenarios to simulate. It has a 70% chance to choose a failed scenario in the past, if any, and 30% chance to randomly choose a scenario allowed by the current phase.

## C.3 ENHANCING MCTS WITH DFS

Algorithm 3 presents the logic for determining whether to unfreeze a frozen transition and, if so, which transition to unfreeze. Our current implementation uses the condition that the reward is negative, which means that MCTS cannot find a model to reach correct decisions for this scenario, and at least one frozen transition is activated (line 2). In our current implementation, MCTS is given enough time to fully explore all state machines relevant to this scenario, so the negative reward is an accurate condition to trigger unfreezing. However, such exhaustive search scales poorly. For better scalability, we may replace this condition with a Z3-style validation to prove that, given the frozen transitions, the model can never reach correct decisions for the corresponding scenario, no matter what other transitions this model includes. We will explore this in the future.

---

**Algorithm 3:** Pseudo code of determining unfreezing

**1 Function** *determine_unfreeze(reward, freeze_list)*:

    /* *Unfreeze when reward is negative and some frozen transition was activated*     */

**2**    **if** *reward* $< 0$ **and** *has_activated(freeze_list)* **then**

        /* *At least one entry can be popped out*     */

**3**        **while** *true* **do**

**4**            *entry* $\leftarrow$ *freeze_list*.pop_one_activated();

**5**            **if** *entry is not fully explored* **then**

**6**                freeze_to_new_value(*entry*);

**7**                *freeze_list*.push(*entry*);

**8**                **Break**;

**9**            **end**

**10**        **end**

**11**    **end**

**12 end**

---

**Algorithm 4:** Pseudo code of determining freezing

**1 Function** *determine_freezing(training_buffer, freeze_list)*:

    /* *Find inputs whose outputs have close probabilities and freeze one*     */

**2**    *tmp* $\leftarrow$ find_ambiguous_inputs(*training_buffer*, *p_min*=0.2, *diff_max*=0.1);

    /* *Sort: later round first, then fewer lost messages first*     */

**3**    *tmp* $\leftarrow$ sort(*tmp*, key=[round_desc, lost_msgs_asc]);

**4**    **if** *tmp* $\neq$ [] **then**

**5**        *cand* $\leftarrow$ *tmp*[0];

**6**        *entry* $\leftarrow$ *cand.freeze_outputs*;

**7**        *freeze_list*.push(*entry*);

**8**    **end**

**9 end**

---

If the condition is met, GGMS unfreezes transitions in the DFS manner. It pops an activated transition from the stack (line 4). If the corresponding input of the transition is not fully explored (line 5), which means that GGMS has not tried to freeze it to all the possible values, GGMS will try to freeze it to a value that has not been explored (line 6) and push this entry back into the stack. Otherwise, GGMS will keep popping until it can find such an entry.

Algorithm 4 presents the logic of determining whether to freeze a new transition and, if so, which one to freeze. Our current implementation uses the heuristics that if multiple outputs for the same input have a probability larger than 0.2 and the difference between their probabilities is within 0.1, then GGMS considers them as targets for freezing (line 2). If multiple such inputs exist, GGMS sorts them based on their round number and the number of lost messages in the input and chooses the one with the highest round number and the lowest number of lost messages as input (lines 3-5). This is based on our experience in the importance of such transitions. Note that this heuristic does not affect the eventual convergence of DFS. Finally, GGMS pushes the newly determined frozen transition into freeze_list.

### C.4 BRUTE-FORCE VALIDATION

Algorithm 5 presents our brute-force algorithm to validate whether the model is fully accurate. It enumerates all the possible scenarios by doing a cross product of all the possible initial state patterns and all the message loss patterns (lines 2-4). Generating all the possible initial state patterns is straightforward: Suppose that there are $N$ processes and $x$ possible initial state values. GGMS enumerates all ways to assign initial states to different processes, generating a total of $x^N$ patterns. Generating all message loss patterns is more complex. GGMS first generates all possible process crash patterns given $phase\_ID$. Then, GGMS generates all message loss patterns based on such crash patterns. In any round before a process crashes, GGMS marks all its messages as not lost.

---

**Algorithm 5:** Pseudo code of validation

---

**1 Function** *validate(phase_ID, model)***:**

   */* Enumerate all patterns for the phase and simulate (can run in parallel)      */*

**2**    *init_state_patterns* ← `gen_all_inputs`(*phase_ID*);

**3**    *msg_loss_patterns* ← `gen_loss_patterns`(*phase_ID*);

**4**    *all_scenarios* ← *init_state_patterns* × *msg_loss_patterns*;

**5**    *failed* ← [];

**6**    **foreach** *scenario* ∈ *all_scenarios* in parallel **do**

**7**      *ok* ← `simulate`(*model, scenario*);

**8**      **if** not *ok* **then**

**9**        *failed*.`append`(*scenario*);

**10**      **end**

**11**    **end**

**12**    **return** *failed*;

**13 end**

---

In any round after a process crashes, GGMS marks all its messages as lost. In the same round as a process crashes, its messages may or may not be lost, and thus GGMS enumerates all such possibilities. Finally, GGMS performs a cross product of the possible message loss patterns of each process.

Then GGMS simulates each of the scenarios to see whether any of them will cause the model to reach incorrect decisions (lines 6-11). Our current implementation parallelizes such simulation of multiple scenarios. As discussed in Section 5, validation is not the bottleneck of our current implementation. In the future, we plan to replace it with a Z3-based validation for better scalability.

### C.5 MONTE-CARLO TREE SEARCH

Algorithm 6 shows the details of the MCTS simulation. At the beginning (line 2), it initializes a new *protocol* object, representing the full protocol execution state with a specific *scenario*. Then it simulates the protocol starting from the current state (line 13). We will describe the `simulate` function in detail later. It returns a probability distribution over transitions for the corresponding state, which is then stored in the training buffer (line 9). Next, it selects a transition based on the simulation results and moves to the next protocol state. This process repeats until the protocol terminates.

The `simulate` function primarily consists of four components. Figure 2 illustrates an example of MCTS: it shows the search tree constructed during the simulation of one episode, and how the path is selected through it. We will introduce these main components in both Algorithm 6 and Figure 2.

- **Selection** (line 17 and line 18). The algorithm will traverse the tree from the root. When selecting a transition from the current node, if a transition is in the `freeze_list`, GGMS selects it directly. Otherwise, GGMS selects a transition based on a balance between exploitation and exploration based on Upper Confidence Bound score as shown in Equation 10. The transition with the highest $U(s, a)$ will be selected in the simulation. $Q(s, a)$ is the accumulative average rewards that taking transition $a$ in state $s$ during simulation. $P(s, a)$ is the probability of choosing transition $a$ in state $s$, given by the policy network. $N(s, a)$ represents the number of times transition $a$ has been chosen in state $s$ during tree search simulations. $c_{puct}$ is a constant parameter that controls the balance between exploitation and exploration. The selection will terminate when the protocol terminates. All visited transitions during selection will be stored.

$$U(s, a) = Q(s, a) + c_{puct} \cdot P(s, a) \cdot \frac{\sqrt{\sum_b N(s, b)}}{1 + N(s, a)} \tag{10}$$

The selection of transitions and message losses follows the same logic but with opposing objectives: the transition selector aims to maximize the final reward, while the message loss selector aims to minimize it. Unlike transition selection, message loss selection relies solely on accumulated rewards without guidance from the network. As shown in Figure 2, within a single iteration

of simulation, the search may follow a path such as C1 → A1 → C3 → A4 (illustrated with solid arrows). In other iterations, different paths may be selected.

Note that, as shown in this algorithm, although each simulation has a targeted scenario, it will explore other reachable scenarios during its search to avoid obviously wrong transitions to other scenarios.

- **Expansion**. When an unexpanded node is reached, all of its available child nodes will be added to the tree for further simulation. For each new node, the visited count and the accumulative reward will be initialized to 0.

- **Evaluation** (line 22). When the *protocol* reaches termination, the final reward is computed based on a predefined reward function (+1 if no correctness property is violated and -1 otherwise). As shown in Figure 2, MCTS selects a particular path in this simulation (C1 → A1 → C3 → A4). At round 1, processes P2 and P3 choose the transition `internal:a(2)`. At round 2, the only living process, P3, selects transition `decision:1(1)`. According to the reward function, this selection results in a final reward of +1.

- **Backup** (line 23). The final reward is backed up along the search path stored in `visited`. The visit count and accumulated rewards of the corresponding nodes are updated accordingly. For example, suppose the selected path in this iteration is C1 → A1 → C3 → A4, and the final reward is +1. In this case, the visit count of all nodes along the path is incremented by 1. The accumulated rewards of nodes A1 and A4 are increased by 1, indicating the protocol has made the right transitions, while those of nodes C1 and C3 are increased by the opposite, −1, indicating the adversary is not able to defeat the protocol. These values will be used to compute the probability of each activated transition at the end of `simulate`.

---

**Algorithm 6:** Pseudo code of Monte-Carlo Tree Search

1 **Function** $run\_mcts$(*scenario, model, freeze_list*)**:**
  /* *protocol: protocol representation*                                         */
2    $protocol \leftarrow$ `initial`(*scenario*);
3    $buffer \leftarrow []$;
4    **while** not $protocol$.*is_done()* **do**
  /* *Run Monte-Carlo tree search from current state*                           */
5      $P \leftarrow$ `simulate`(*protocol, model, freeze_list*);
  /* *Pick transition based on simulation results*                              */
6      $transition \leftarrow$ `select`($P$);
  /* *Advance to next round*                                                     */
7      $protocol$.`step`(*transition, scenario*);
  /* *Non-zero reward only at the last round*                                    */
8      $reward \leftarrow$ `get_reward`();
  /* *Record (state, simulated probabilities)*                                   */
9      $buffer$.`append`($\langle$`current_state`(), $P\rangle$);
10    **end**
11    **return** $buffer, reward$;
12 **end**
13 **Function** `simulate`(*protocol, model, freeze_list*)**:**
14    **for** $i \leftarrow 0$ **to** $iter$ **do**
  /* *Store visited path during simulation*                                      */
15      $visited \leftarrow []$;
16      **while** $protocol$ is not done **do**
  /* *Select transitions and message losses based on Equation 10*                */
17        $transition \leftarrow$ `select_transition`(*model, freeze_list*);
18        $loss \leftarrow$ `select_message_loss`();
19        $protocol$.`next`(*transition, loss*);
20        $visited$.`append`(*transition, loss*);
21      **end**
22      $reward \leftarrow$ `get_reward`();
23      `backup`(*reward, visited*);
24    **end**
25 **end**

---

## D EXAMPLES OF SUPERPOSITION PROBLEM IN MCTS

We use the consensus protocol as an example to illustrate how superposition, that is, combining transitions from multiple correct versions of the protocol can occur and affect the convergence of MCTS. Keep in mind that consensus requires that 1) every process makes the same decision, and 2) if the initial input to every process is init:0, then the decision must be decision:0; if the initial input to every process is init:1, then the decision must be decision:1; if some processes has init:0 as the input and some have init:1, then the decision could be either decision:0 or decision:1.

Human knowledge tells that, inside such a protocol, each process should use an internal state to record its intention, and multiple processes should exchange their intentions to resolve divergence among processes. For example, if a process observes that all processes have init:0 in the first round, it may change its internal state to internal:a, indicating that it intends to go to decision 0. Note that if a process observes some init:0 and some Lost, it should transit to internal:a as well, since Lost may be from a process with init:0. Similarly, if a process observes init:1 or Lost, but no init:0, then it can transit to internal:b, indicating that it intends to go to decision 1. However, if a process observes both init:0 and init:1, it can transit to either internal:a or internal:b. Processes can exchange such intentions for multiple rounds until a consensus can be reached.

There are at least two reasons for the existence of multiple versions of the correct protocols. First, without human knowledge, the protocol may assign certain meanings to arbitrary internal states, creating multiple equivalent protocols. For example, while the above example uses internal:a for intention decision:0 and internal:b for intention decision:1, we can swap this mapping to create an equivalent protocol. This creates a problem for MCTS. When MCTS simulates the scenario with all processes having init:0 as input; it may find that it is feasible for a process to transit to internal:a in this case, and finally transit to decision:0. When MCTS simulates the scenario with all processes having init:1 as input; it may also find that it is feasible for a process to transit to internal:a in this case, and finally transit to decision:1. Although the solution to each individual scenario is correct, combining them is incorrect, since we should not let the all init:0 scenario and the all init:1 scenario transit to the same internal state, as there is no way to distinguish them in the later rounds.

The second reason comes from the inherent ambiguity allowed by the protocol, that is, if some processes have init:0 as input and some have init:1, then the decision could be either decision:0 or decision:1. To give a concrete example about how this causes problems for MCTS, suppose that there are three processes participating in this protocol. Their input states are [init:0, init:0, init:1]. Suppose Process 0 crashes in the first round; its message is received by Process 1, but not Process 2 (Scenario 1). So Process 1's input is [init:0, init:0, init:1] (Input A) and Process'2 input is [Lost, init:0, init:1] (Input B). Due to the ambiguity of the initial input, we know that Input A and Input B can transit to either internal:a or internal:b, as long as they transit to the same internal state. If MCTS only simulates this scenario, it may find this constraint and thus give $[A \rightarrow$ internal:a$]$ and $[B \rightarrow$ internal:a$]$ a higher probability than $[A \rightarrow$ internal:b$]$ and $[B \rightarrow$ internal:b$]$. However, Input A and Input B may appear separately in other scenarios as well. For example, if Process 0 does not crash in the above example, all processes will have Input A but no Input B (Scenario 2); if Process 0 crashes and both Process 1 and Process 2 miss its message, then both will have Input B but no Input A (Scenario 3). Since MCTS simulates each of these scenarios independently, it may end up preferring $[A \rightarrow$ internal:a$]$ in Scenario 2 and preferring $[B \rightarrow$ internal:b$]$ in Scenario 3. And when MCTS combines the probabilities from all scenarios, it may let A and B transit to different internal states. Again, in this example, MCTS finds a correct solution for each scenario, but it is incorrect to combine those solutions.

## E MODEL SELECTION AND HYPERPARAMETERS

For model selection, we use a Transformer model, which has become popular in language modeling tasks. The architecture of this model is presented in Table S3. The Transformer model includes only the encoder component of the standard Transformer architecture, as the task does not require

| Layer Type | Input Shape | Output Shape |
|---|---|---|
| One-hot Encoder | (input) | (input, encode_dim) |
| Transformer Encoder | (input, encode_dim) | (input, hidden) |
| GlobalAveragePooling1D | (input, hidden) | (hidden) |
| Output | (hidden) | (output) |

Table S3: The Transformer architecture used to learn the state machine

| Layer Type | Size | Output |
|---|---|---|
| FC-1 | 3x128 | 1x128 |
| FC-2 | 128x64 | 1x64 |
| FC-3 | 64x32 | 1x32 |
| Output | 32x3 | 1x3 |

Table S4: The MLP architecture used to learn the state machine

contextual information like translation tasks. Additionally, a one-hot encoding layer is used at the input to transform categorical values into unique vectors. The Transformer encoder outputs a tensor of shape batch $\times$ input_length $\times$ hidden_dim. We apply a global pooling layer to aggregate the outputs across the input sequence into a vector of shape batch $\times$ hidden_dim, which is then passed through an output layer to obtain the final prediction vector. We also tried the MLP model. Table S4 demonstrates the MLP model architecture that we use. It contains three fully connected layers (FC) and one output layer. We use ReLU as the activation function after each FC layer. We use cross-entropy as the loss function for both models.

## F  EXPERIMENT SETTING

We run all experiments on CloudLab. The server we use is equipped with a 16-core AMD 7302P CPU running at 3.00GHz and has 128GB of memory. We implement the system in Python and use Keras for model training. The learning rate is set to 0.001, and the cross-entropy loss function is used to update the model parameters. In our experiments, we simulate 100 scenarios in each episode. For each simulation, the number of iterations is determined by the scale of the setting, calculated as #rounds $\times$ #processes $\times$ 1000. We freeze a new transition every 5 episodes to ensure that all previously frozen transitions have propagated. To detect training failure, we set a timeout for each run based on the scale. Specifically, we allocate 1 day for 2-process settings, 2 days for 3-process settings, and 4 days for 4-process settings. Some distributed protocols require all processes to make the same decision, so a process may not need the process ID as the input, since processes with the same input, regardless of the process ID, should transit to the same state. We find that this is true for both the atomic commit protocol and the consensus protocol we have investigated, so we disabled the process ID in our experiments to reduce search space.

## G  ADDITIONAL EVALUATION RESULTS

Figure S1 shows the number of counterexamples found by our verifier in one setting. At the beginning, the number of counterexamples is low, since guided MCTS focuses on some specific scenarios. Then, after relaxation, guided MCTS starts to explore more scenarios, leading to more counterexamples. GGMS performs two rounds of freezing, and the number of counterexamples drops to zero. Finally, after more relaxation, GGMS does not find more counterexamples.

Figure S2 presents another example illustrating the number of counterexamples found by our verifier under one setting. GGMS first freezes a transition. It then discovers that, under this frozen condition, no correct model can be learned. Consequently, GGMS unfreezes the transition and continues training. The system eventually learns a correct model, though it requires more time compared to the example shown in Figure S1.

Our current implementation uses the Transformer model as discussed in Section E. We also tried the MLP model. For the following settings ac-2-1, con-2-1, ac-3-2, and con-3-2, the success rate of using the MLP model is 100%, 70%, 10%, and 20%, respectively, compared to 100%, 100%, 60%, and 100% of using the Transformer model.

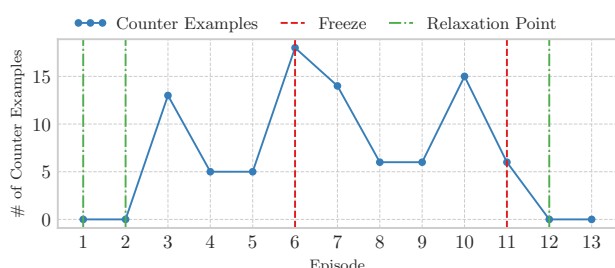

Figure S1: Number of counter examples for ac-3-2.

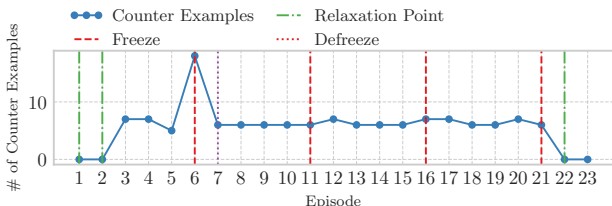

Figure S2: Number of counter examples for ac-3-2.

## H    COMPETING APPROACHES

We briefly position GGMS among several related families of methods, focusing on differences in goals and outcomes.

**Classical distributed protocols and verification.**    Classical distributed protocols such as two-phase commit and atomic commit (Gray, 1978; 2005; Corbett et al., 2012; Zhang et al., 2015), Paxos and its variants (Lamport, 2001; 1998; Lamport et al., 1982; Moraru et al., 2013; Ongaro & Ousterhout, 2014; Castro & Liskov, 1999; Kotla et al., 2007; Giridharan et al., 2024) are hand-designed by experts and then validated by formal proofs or model checking. Frameworks such as IronFleet, I4, DistAI, and Basilisk (Hawblitzel et al., 2015; Ma et al., 2019; Yao et al., 2021; Zhang et al., 2025b) start from a human-written protocol and focus on proving its correctness, often for arbitrarily many processes. In contrast, GGMS treats the protocol itself as the object of synthesis: starting only from a specification of safety properties and a fixed number of processes, it searches in the space of global state machines and uses exhaustive model checking to ensure that the learned protocol satisfies the specification. Thus, classical work assumes the protocol and proves properties, whereas GGMS automates protocol construction and then certifies the resulting state machine (for the chosen bound).

**Self-play and learning in multi-agent systems.**    Self-play deep reinforcement learning with tree search has achieved superhuman performance in perfect- and imperfect-information games such as backgammon, Atari, Go, poker, and Hanabi (Tesauro et al., 1995; Mnih et al., 2013; 2015; Silver et al., 2017; Lample & Chaplot, 2017; Wan et al., 2018; Tan et al., 2019; Li et al., 2020; Bard et al., 2020; Brown & Sandholm, 2019a; Anthony et al., 2017; Brown & Sandholm, 2019b; Sudhakar et al., 2025). These methods aim to maximize expected reward and typically output a neural policy that is not formally verified. GGMS borrows Monte Carlo Tree Search and self-play as search and data-generation tools, but with a different objective: instead of winning a game on average, it must synthesize a protocol that satisfies strict safety properties under worst-case crashes and message losses.

Closer to our setting, Khanchandani et al. (2021) use self-play to learn algorithms for a distributed directory problem, and verification-guided multi-agent RL approaches (e.g., Zhang et al. (2025a); Belardinelli et al. (2024)) combine learning with model checking to enforce temporal-logic specifications. These methods learn policies or algorithms and typically use model checking as a verifica-

tion step or shaping signal, with correctness evaluated on sampled executions or specific scenarios. GGMS instead searches directly in a symbolic space of global state machines for asynchronous crash-prone message-passing systems, and uses model checking as a hard oracle: any violating protocol is rejected, and counterexamples are fed back into training until no counterexamples remain within the explored state space. In this sense, GGMS is closer to protocol synthesis with built-in verification than to performance-driven policy learning.

**Automata learning and program synthesis.** The global state machine representation in GGMS is reminiscent of deterministic finite automata (DFA). Classical DFA learning methods (de la Higuera, 2005; Heule & Verwer, 2010) infer an automaton consistent with a fixed set of positive and negative example traces. GGMS also produces a finite-state machine, but its data is not a static dataset: executions are generated online via MCTS-guided self-play and iteratively refined using counterexamples from model checking, and the target is satisfaction of distributed safety properties under all crash and message-loss patterns modeled by the verifier.

At a higher level, GGMS is also related to classical program synthesis, particularly counterexample-guided inductive synthesis (CEGIS) as introduced in program sketching (Solar-Lezama, 2008). In CEGIS, a synthesizer proposes candidate programs from a constrained search space, and a verifier either accepts the candidate or returns a counterexample input; the counterexample is added to the training set and the loop repeats. GGMS follows a similar high-level architecture: a search procedure proposes candidate protocols and a model checker either accepts them or returns offending executions. However, typical CEGIS systems operate on general-purpose sequential programs (often using SMT-based search) and reason about input/output behavior, whereas GGMS searches over a structured space of distributed protocol state machines under asynchronous semantics and crash/message-loss faults, guided by reinforcement learning and MCTS rather than purely symbolic search. Moreover, GGMS explicitly explores the global state space of the protocol (up to a bound) to guarantee correctness for all executions within that model.

**Large-scale reasoning and coding agents.** Recent large-scale systems also combine powerful search or reinforcement learning with formal environments. AlphaProof is an AlphaZero-inspired agent for formal mathematics that operates inside the Lean theorem prover (Hubert et al., 2025). Each problem instance is a formal theorem; the agent observes proof states, proposes tactics, and uses MCTS guided by a transformer "proof network" to search for proofs, with the Lean kernel checking correctness. The goal is to *find proofs* of given statements within a fixed formal system, and the outcome is a proof term.

AlphaEvolve is an evolutionary coding framework for scientific and algorithmic discovery (Novikov et al., 2025), in which large language models propose edits to candidate programs that are then executed and scored by problem-specific evaluation functions. The framework has been applied to discover new algorithms and improve implementations in several domains. In AlphaEvolve, candidate solutions are arbitrary programs, and their correctness or utility is judged by external evaluation metrics defined by the user.

GGMS is complementary to both: rather than proving theorems in a general-purpose proof assistant or evolving arbitrary code under user-defined metrics, GGMS operates in a fixed asynchronous message-passing model with crash and message-loss faults, searches over a constrained symbolic space of distributed protocols, and uses exhaustive model checking of all executions within this model to certify correctness (for a fixed number of processes).

## I    EXPERIENCE WITH GPT

We tested GPT 5.1 with extended thinking to see whether it can accomplish a similar goal. Concretely, we use a prompt to describe the requirements and ask GPT to generate a protocol; we manually read the protocol generated by GPT and provide a counterexample if any; we repeat this until GPT either gets a correct protocol or concludes that it is impossible.

For the consensus protocol, GPT finds one correct version in one shot, by correctly pointing to the FloodSet algorithm. The transcript is here: `https://chatgpt.com/share/69164ba6-11d4-800a-8acc-29c061b7d8f7`. Since our learned algorithm is slightly different from FloodSet (FloodSet maintains the set of proposals during the protocol and chooses the

min at the end; our learned algorithm applies min or other functions during the protocol, which saves some space), we asked GPT whether it can optimize the protocol. GPT then gets a protocol that is similar to our learned one (maybe with a different decision function).

For the atomic commit protocol, our experience is different, probably because we change the problem definition to some extent: Existing atomic commit protocols like two-phase commit (2PC) target asynchronous networking environment, but ours targets synchronous networking environment, and we are not aware of an existing atomic commit protocol targeting synchronous networking environment. The transcript is here: `https://chatgpt.com/share/6916864b-8e90-800a-ba99-57b0096aef6d`.

In the first iteration, GPT outputs a protocol similar to FlootSet, using '0' to represent initial proposal "abort" and '1' to represent initial proposal "commit". However, this protocol does not work if a process proposes "abort" but fails to send any message before it crashes. In this case, atomic commit still requires the protocol to decide abort, but FloodSet may decide to commit if all other processes propose "commit". We presented this counterexample to GPT. GPT argued that this may be due to a misunderstanding of whether a process should count if it crashes before it sends out anything. We clarified to GPT that such a process should still count as an atomic commit.

In the second iteration, GPT outputs a slightly modified FloodSet algorithm, which treats both initial proposal "abort" and a lost message as '0' and still treats initial proposal "commit" as 1. While this algorithm addresses the above counterexample, it introduces a new counterexample that if all processes have state "commit" at the end of round $f$, but one process crashes in round $f + 1$, then processes may make different decisions. We gave this counterexample to GPT.

After an extended chain-of-thought process, the model concluded that designing such a protocol was **impossible** under the given constraints. From analyzing the thinking summaries, we found that viable ideas were considered but ultimately abandoned. Concretely, a correct protocol is to treat "commit" as '1' and treat "abort" and "message loss" as '0' in the first round; it then can ignore "message loss" in later rounds and run the standard FloodSet algorithm. This is because as long as one process stays at '1' in the first round, that means that all processes' initial proposals must be "commit", so it is fine to run a standard consensus afterwards. In its thinking log, we found that GPT is considering ignoring "message loss" in round $f + 1$, but not in earlier rounds. This is, of course, still not correct, but it is heading towards the right solution; however, GPT gave up.

These results suggest that while LLMs can parse local constraints, they struggle to maintain the global coherence required for distributed protocols, even in relatively simple settings where GGMS succeeds.