# OpenReview forum: "Learning Provably Correct Distributed Protocols Without Human Knowledge"
_ICLR.cc/2026/Conference — Submitted to ICLR 2026_

### Official Review · Reviewer_gi5q · 2025-10-30

**Soundness:** 2
**Presentation:** 2
**Contribution:** 2
**Rating:** 4
**Confidence:** 2

**Summary:**

This paper introduces GGMS (Guided Global Monte Carlo Search), a framework for automatically synthesizing provably correct distributed protocols without relying on human-designed examples. The approach integrates Monte Carlo Tree Search (MCTS), Depth-First Search (DFS), and model checking, iteratively exploring protocol candidates and verifying their correctness through formal verification tools. The authors demonstrate GGMS on small-scale atomic commit and consensus tasks, showing that it finds correct protocols more reliably than baseline approaches such as standard MCTS or MCTS+DFS.

I should note that I am not an expert in distributed systems or formal verification, and my evaluation is therefore based on a non-specialist perspective. My review focuses on clarity, accessibility, and the apparent coherence and novelty of the approach, rather than on detailed technical verification.

**Strengths:**

- **Ambitious and conceptually strong idea.** Automating the synthesis of provably correct distributed protocols is a highly ambitious and forward-looking goal. While foundational protocols such as Paxos already exist and are widely implemented, the authors target a different problem: automating the design and verification of such protocols. In practice, even small variations in assumptions (e.g., timing, fault tolerance, message loss) require new, carefully reasoned protocols. Automating this process could reduce the time and human expertise needed to design correct distributed systems, which justifies the research direction despite the existence of established solutions like Paxos.

- **Clear motivation.** The introduction effectively conveys why distributed protocol design is difficult and how formal verification guarantees robustness. The analogy to AlphaGo-style learning is intuitive, though distributed protocols require perfect correctness rather than probabilistic success.

- **Methodologically coherent.** The integration of MCTS, DFS, and model checking appears consistent and well justified. The paper’s internal logic is clear, and limitations are discussed transparently.

- **Honest discussion of limitations.** The authors openly acknowledge the synchronous-network assumption, limited scale, and prototype nature of their implementation.

- **Potentially broad implications.** If extended successfully, this line of work could influence future research on verified ML and the automation of safe system design.

**Weaknesses:**

- **Accessibility.** Section 3 (Modeling a Distributed Protocol with Zero Knowledge) is difficult to follow for non-experts. The abundance of formal states and transitions interrupts readability. A more intuitive overview before the formalism would help.

- **Figures.** Figures 1–4 are hard to read due to small font sizes and dense labeling. Redrawing them with larger fonts and clearer legends would improve accessibility.

- **Empirical scope.** Evaluation is restricted to small-scale examples (atomic commit, consensus). It remains unclear whether GGMS could handle larger or asynchronous settings.

- **Nature of the contribution.** It is somewhat unclear whether the paper’s emphasis is theoretical or empirical (demonstration on simple tasks). The small-scale experiments and my lack of experise make it hard to assess which aspect dominates.

- **Background and related work overlap.** The long background and short related-work sections could be merged for better flow and a unified perspective.

- **Future work section.** Listing implementation upgrades (rewriting in C, GPU acceleration) feels too engineering-focused. A more research-oriented outlook—e.g., scaling to asynchronous or Byzantine models—would be stronger.

**Questions:**

- Could the authors clarify where exactly the claim “freezing an entire MCTS path does not cause missing of correct protocols” is proven or justified? Is this statement formally captured, or should the reader understand it as an empirical observation?

- On page 7, the paper mentions difficulties in “propagating the effects of DFS freezes through MCTS” and refers to the need for sufficient “propagation power.” Could the authors please clarify what is meant by propagation power in this context. Additionally, would the use of set-invariant or permutation-invariant architectures (e.g., Set Transformers) potentially help enforce more consistent transitions across related states?

- Does GGMS rediscover existing protocols such as 2PC or produce novel but functionally equivalent variants? Would it be expected of GGMS to find exisiting protocols?

- How does the algorithm scale in general?

- All experiments are run on CPUs using a single-threaded Python implementation. Could the authors clarify whether this choice reflects a limitation of the GGMS algorithm (e.g., due to sequential dependencies in MCTS and model checking), or if it was simply an implementation decision? Do you expect the method to benefit from GPU acceleration or distributed parallelization in future versions?

---

> ### Author Response · Authors · 2025-11-21
>
> **Weaknesses:**
>
> * **Accessibility.** Section 3 (Modeling a Distributed Protocol with Zero Knowledge) is difficult to follow for non-experts.
>   **Response**: We have added an intuitive example in Section 2.1.
> * **Figures.** Figures 1–4 are hard to read due to small font sizes and dense labeling.
>
>   **Response**: We have revised Figures 1 and 2 and replaced Figures 3 and 4 with a summary table.
> * **Empirical scope.** Evaluation is restricted to small-scale examples (atomic commit, consensus).
>   **Response**: Please see the \[Scalability\] discussion at the beginning.
> * **Nature of the contribution.** It is somewhat unclear whether the paper’s emphasis is theoretical or empirical (demonstration on simple tasks). The small-scale experiments and my lack of experise make it hard to assess which aspect dominates.
>   **Response**: Our primary contribution is algorithmic/methodological rather than purely theoretical or purely empirical. The theoretical/formal part of the work is the precise modeling and the use of exhaustive model checking to guarantee that any protocol output by GGMS is correct for all executions up to the chosen bound. The empirical component is deliberately small-scale and illustrative: the tasks are chosen so that (a) the entire state space can be explored, (b) we can exhaustively verify correctness, and (c) the learned protocols are interpretable. The experiments are meant to demonstrate that GGMS can in fact synthesize nontrivial distributed protocols under these constraints and that its components (guided MCTS, DFS, counterexample feedback) meaningfully affect the outcome, rather than to provide large-scale performance benchmarks.
>   We have revised the introduction to make the nature of the contribution clearer.
> * **Background and related work overlap.** The long background and short related-work sections could be merged for better flow and a unified perspective.
>
>     **Response**: We have extended the discussion about related work as suggested by reviewer **nn79.**
>
> * **Future work section.** Listing implementation upgrades (rewriting in C, GPU acceleration) feels too engineering-focused. A more research-oriented outlook—e.g., scaling to asynchronous or Byzantine models—would be stronger.
>
>  	**Response**: We discussed asynchronous and Byzantine models in the future work section. We moved the implementation improvement to the scalability discussion, where it is more relevant.
>
> **Questions:**
>
> * Could the authors clarify where exactly the claim “freezing an entire MCTS path does not cause missing of correct protocols” is proven or justified? Is this statement formally captured, or should the reader understand it as an empirical observation?
>   **Response**: This is formally captured in Theorem 2, which is proved in Appendix B.
> * Could the authors please clarify what is meant by propagation power in this context. Additionally, would the use of set-invariant or permutation-invariant architectures (e.g., Set Transformers) potentially help enforce more consistent transitions across related states?
>   **Response**: (propagation power) By “propagation power,” we refer to the strength of the learning signal required to align unfrozen states with frozen ones. It measures how effectively the MCTS simulations can generalize a hard constraint (imposed by a DFS freeze) to related but visited states (e.g., symmetric states or subsets), thereby propagating the correct decision logic across the state space without requiring explicit freezes for every variation. We clarified near the bottom of page 2\.
>   (permutation invariance) We leverage exactly this principle in our current architecture. As detailed in Appendix E, our policy network utilizes a Transformer encoder followed by a global pooling layer. This architecture renders the model permutation-invariant with respect to the order of incoming messages.
> * Does GGMS rediscover existing protocols such as 2PC or produce novel but functionally equivalent variants?
>   **Response**: We have added a discussion at the end of evaluation. In short, GGMS rediscovers an existing consensus protocol, but for atomic commit, since we change the problem definition to some extent, GGMS finds a new protocol as far as we know. Concretely, 2PC targets asynchronous networking and our protocol targets synchronous networking.
> * How does the algorithm scale in general?
>   **Response**: Please see the \[Scalability\] discussion at the beginning.
> * All experiments are run on CPUs using a single-threaded Python implementation.
>
>   **Response**: It is simply an implementation decision. MCTS is naturally well parallelizable and AlphaGo implements a distributed version. We do not see any reason we cannot do the same. Z3 is known to be not very parallelizable.

---

> > ### Comment · Reviewer_gi5q · 2025-11-27
> >
> > ## Post-rebuttal comments
> >
> > I thank the authors for the responses and revisions.
> >
> > Several of my earlier concerns have been at least partially addressed. In particular,
> > - the added intuitive example in Section 2.1 and the revised figures / summary table appear to improve accessibility,
> > - the authors now explicitly point to Theorem 2 and Appendix B as the place where the claim about freezing an entire MCTS path is formalized, and they clarify the notion of “propagation power” and the role of permutation invariance in the architecture, and
> > - the new discussion at the end of the evaluation clarifies to what extent GGMS rediscovers known protocols versus finding new ones, while the expanded scalability and future-work discussion helps to position the contribution more clearly.
> >
> > However, the empirical evaluation remains restricted to relatively small, synchronous examples, and questions about scalability and applicability to more realistic settings (e.g., larger systems, asynchronous or Byzantine models) are still only partially answered at a conceptual level. Overall, I continue to see the work primarily as a promising prototype with limited empirical scope.
> >
> > Regarding venue fit, I do believe that automating the design of critical systems via machine learning and formal methods is an important direction that should be represented at the conference. That said, given the current presentation and the prototype nature of the evaluation, I am not fully convinced that this particular version of the work is ready for inclusion.

---

### Official Review · Reviewer_EcFw · 2025-10-31

**Soundness:** 3
**Presentation:** 2
**Contribution:** 3
**Rating:** 6
**Confidence:** 3

**Summary:**

This paper presents GGMS, a learning framework for automatically designing provably correct distributed protocols without human knowledge. The authors formulate distributed protocol design as a search problem in a game with imperfect information, where processes must coordinate to reach consensus despite partial observability, message losses, and potential crashes. GGMS combines a specialized Monte Carlo Tree Search variant inspired by Alpha-Zero with a transformer-based action encoder, global depth-first search to escape local minima, and iterative feedback from a model checker to guarantee correctness properties specified in SMT. Authors include experimental validation on synchronous atomic commit and consensus protocols.

**Strengths:**

I believe the authors tackle a challenging problem, automatically learning provably correct distributed protocols without human knowledge. I found quite intuitive the formulation of protocol design as a search problem in an imperfect information game. However I am not familiar with these protocols and I cannot comment on whether this is the first work attempting that. Nevertheless, The GGMS framework introduces several innovations that go beyond traditional MCTS including the integration of global depth-first search (DFS) to escape local minima while maintaining theoretical guarantees of eventual convergence. The authors also introduce a "Guided MCTS" component, which uses counterexamples from the model checker to bias the search toward problematic scenarios.

In general the paper gives a sound theoretical foundation given the overall assumptions and the experimental results show the gains with respect vanilla MCTS

**Weaknesses:**

The paper makes several restrictive assumptions that significantly limit its applicability to real-world distributed protocols -although clearly highlighted by the authors-: 1) The method is restricted to synchronous networks 2) GGMS assumes that processes can only send identical messages to all other processes (rather than different messages to different recipients) 3) The usage of model-checking, which is computationally exhaustive, restricts the number of processes that can be checked.  The experimental results confirm severe scalability limitations - the largest evaluated setting involves only 4 processes with 2 failures. The exponential growth in running time (Figure 4) and the dominance of brute-force model checking in the verification phase suggest the approach will struggle with realistic protocol sizes. The authors mention switching to Z3 for verification but provide no evidence this would resolve the scalability bottleneck.

The paper's introduction assumes significant background knowledge in distributed systems, making it harder to a general ML audience like ICLR. The first two pages dive directly into technical concepts like "atomic commit," "consensus," and "Byzantine Fault Tolerance" without providing intuitive explanations or toy examples. A simple running example (e.g., 3 processes trying to agree on a value with 1 possible failure) introduced early would greatly improve accessibility. The formal problem definition in Section 2 jumps into notation-heavy descriptions without first providing intuition about what a distributed protocol actually does.

I also found the empirical evaluation somewhat limited with it being  restricted to only two protocol types (atomic commit and consensus) in their simplest forms. I also missed ablations on the different modifications to MCTS that authors introduced.

**Questions:**

I don't have particular questions but I want to conclude that despite the restrictive assumptions, these don't hinder the contributions. I see this more as a grounding work and I don't see these assumptions as reason for rejection. I do think that including running examples on the introduction and ablations would improve the clarity and soundness of the paper.

---

> ### Author Response · Authors · 2025-11-21
>
> **Weaknesses:**
> The paper makes several restrictive assumptions that significantly limit its applicability to real-world distributed protocols \-although clearly highlighted by the authors-: 1\) The method is restricted to synchronous networks 2\) GGMS assumes that processes can only send identical messages to all other processes (rather than different messages to different recipients) 3\) The usage of model-checking, which is computationally exhaustive, restricts the number of processes that can be checked. The experimental results confirm severe scalability limitations \- the largest evaluated setting involves only 4 processes with 2 failures. The exponential growth in running time (Figure 4\) and the dominance of brute-force model checking in the verification phase suggest the approach will struggle with realistic protocol sizes. The authors mention switching to Z3 for verification but provide no evidence this would resolve the scalability bottleneck.
>
> **Response**: To clarify, running time is NOT dominated by brute-force model checking but by MCTS, and that’s why we are OK with brute-force model checking now. Please also see the \[Scalability\] discussion at the beginning.
>
> The paper's introduction assumes significant background knowledge in distributed systems, making it harder to a general ML audience like ICLR. The first two pages dive directly into technical concepts like "atomic commit," "consensus," and "Byzantine Fault Tolerance" without providing intuitive explanations or toy examples. A simple running example (e.g., 3 processes trying to agree on a value with 1 possible failure) introduced early would greatly improve accessibility. The formal problem definition in Section 2 jumps into notation-heavy descriptions without first providing intuition about what a distributed protocol actually does.
>
> **Response**: We have added a brief example in Section 1 (second paragraph) and expanded the example with more details (but still with intuition) in Section 2.1. Hope that helps.
>
> I also found the empirical evaluation somewhat limited with it being restricted to only two protocol types (atomic commit and consensus) in their simplest forms. I also missed ablations on the different modifications to MCTS that authors introduced.
>
> **Response**: We agree with the limitation “only two protocols in their simplest forms”. We are working on a version with early decisions, suggested by reviewer nn79. Before the submission, we have also tried the distributed locking protocol, but since it is even simpler, we did not include it in the submission.
>
> For ablations, “MCTS” includes only techniques described in Section 4.1. “MCTS+DFS” includes Section 4.1 and Section 4.2. “GGMS”, which is the full version, includes Sections 4.1, 4.2, and 4.3. We have clarified them in the text.
>
> **Questions:**
> I don't have particular questions but I want to conclude that despite the restrictive assumptions, these don't hinder the contributions. I see this more as a grounding work and I don't see these assumptions as reason for rejection. I do think that including running examples on the introduction and ablations would improve the clarity and soundness of the paper.
>
> **Response**: As discussed above, we have added examples in the revision. We explained ablations in Section 5 (before “success rate”).

---

### Official Review · Reviewer_nn79 · 2025-11-01

**Soundness:** 3
**Presentation:** 3
**Contribution:** 3
**Rating:** 4
**Confidence:** 4

**Summary:**

The paper proposes GGMS, a framework to learn distributed protocols that are provably correct, using only the desired properties and no human protocol knowledge. The approach models each process as a deterministic state machine and uses: (1) MCTS guided by a transformer policy, (2) a global DFS that “freezes” ambiguous transitions to break out of local minima, and (3) repeated feedback from a brute-force model checker that provides counterexamples for further training. The authors argue GGMS will not miss correct solutions under mild assumptions and show experiments on synchronous atomic commit and consensus, where GGMS achieves higher success rates than MCTS or MCTS+DFS within time limits, or small numbers of rounds.

**Strengths:**

1. The paper introduces a new integration of guided MCTS, DFS freezing, and model-checking feedback for protocol synthesis. It provides theoretical results ensuring that freezing does not exclude correct solutions.

2. The clarity is good. Assumptions are stated (synchronous, same state machine per process, last-round decisions). Pseudo-code and examples make the method understandable. The modeling of processes as deterministic state machines, the learning–verification loop, and counterexample-guided retraining are clearly explained.

3. Reproducibility is strong: code is included, and the settings are well-documented.

**Weaknesses:**

1. GGMS assumes synchronous communication, identical deterministic state machines, and decisions in the final round. These simplifications are standard in this line of research. Still, showing one partially relaxed setting (e.g., I guess early decision or message subset) would strengthen the message that GGMS generalizes beyond toy cases.

---

2. The empirical section is minimal. Only a few experiments are tested, with few figures and no summary tables for success rates, runtime, or verification outcomes. The results are presented in qualitative line plots without statistical analysis or detailed metrics. More quantitative evidence or tabulated comparisons would make the evaluation more convincing.

---

3. Baseline comparisons are not strong enough: only MCTS and MCTS+DFS are tested, with no comparison to prior methods. Even a brief qualitative discussion or positioning would help readers place GGMS among existing work. This is my main concern. In addition, the analysis has limited insight: it mostly shows that GGMS, an intuitively promising integration of several modules, outperforms ablations that remove one or two modules.

---

4. The theoretical “provably correct” claim seems too strong. The two theorems are logically coherent within a finite-search setting (finite search space, accurate unfreezing), but they do not provide strong guarantees for practical learning or correctness, especially when stochasticity is introduced. The claim of “eventual convergence” could still imply astronomically long runtimes, as there is no complexity bound.

---

5. Minor issue: The figures could be improved. Figures 1–5 lack detailed captions, which makes them hard to understand.

**Questions:**

I don’t have many specific questions; my main concern is the experiments and insights behind them. Could the authors add qualitative comparisons with prior work, not only MCTS or MCTS+DFS, to clarify how GGMS differs in goals and outcomes?

---

> ### Author Response · Authors · 2025-11-21
>
> **Weaknesses:**
>
> 1. GGMS assumes synchronous communication, identical deterministic state machines, and decisions in the final round. These simplifications are standard in this line of research. Still, showing one partially relaxed setting (e.g., I guess early decision or message subset) would strengthen the message that GGMS generalizes beyond toy cases
>    **Response**: We are working on a version to support early decision..
>
> ---
>
> 2. The empirical section is minimal. Only a few experiments are tested, with few figures and no summary tables for success rates, runtime, or verification outcomes. The results are presented in qualitative line plots without statistical analysis or detailed metrics. More quantitative evidence or tabulated comparisons would make the evaluation more convincing.
>    **Response**: We added a summary table for success rates and running time in evaluation (replaced Figures 3 and 4). GGMS only outputs a protocol if it passes verification.
>
> ---
>
> 3. Baseline comparisons are not strong enough: only MCTS and MCTS+DFS are tested, with no comparison to prior methods. Even a brief qualitative discussion or positioning would help readers place GGMS among existing work. This is my main concern. In addition, the analysis has limited insight: it mostly shows that GGMS, an intuitively promising integration of several modules, outperforms ablations that remove one or two modules.
>
>   	**Response**: For positioning, please see the answer in the “Questions” section.
>
> 	For “limited insight”, we have added a discussion about why MCTS and MCTS+DFS are not sufficient in our experience at the end of “success rate”. Any further suggestions are welcome.
>
> ---
>
> 4. The theoretical “provably correct” claim seems too strong. The two theorems are logically coherent within a finite-search setting (finite search space, accurate unfreezing), but they do not provide strong guarantees for practical learning or correctness, especially when stochasticity is introduced. The claim of “eventual convergence” could still imply astronomically long runtimes, as there is no complexity bound.
>    **Response**: There is some confusion here. There are two kinds of proofs. The “provably correct” term in the title means that the resulting model we train is provably correct. This is achieved through verification (brute force for now and maybe Z3 or model checker in the future). These methods are quite widely used in the distributed systems community. The two theorems in the paper try to prove that our training process can eventually find a correct model, if it exists. These two theorems indeed are based on certain assumptions and do not have a complexity bound.
>
> ---
>
> 5. Minor issue: The figures could be improved. Figures 1–5 lack detailed captions, which makes them hard to understand.
>    **Response**: We have revised Figures 1 and 2\. We replaced Figures 3 and 4 with the new summary table for better readability. We do not have Figure 5\.
>
> **Questions:**
> I don’t have many specific questions; my main concern is the experiments and insights behind them. Could the authors add qualitative comparisons with prior work, not only MCTS or MCTS+DFS, to clarify how GGMS differs in goals and outcomes?
>
> **Response**: In the revised manuscript, we have made changes specifically aimed at helping readers understand how GGMS relates to prior methods beyond the MCTS and MCTS+DFS baselines. The Related Work section now explicitly highlights that GGMS differs in both its goal and its outcome and refers to a new appendix section, “Competing Approaches”, that provides the qualitative comparisons the reviewer requested. There we position GGMS with respect to: classical hand-designed protocols and verification frameworks, self-play RL and verification-guided multi-agent learning, automata learning and CEGIS-style program synthesis, and large-scale reasoning/coding agents such as AlphaProof and AlphaEvolve. For each family, we explicitly describe how its goals and outputs differ from those of GGMS.
>
> We hope these additions address the reviewer’s main concern by providing the requested qualitative positioning of GGMS among existing work.

---

### Official Review · Reviewer_ms1M · 2025-11-03

**Soundness:** 2
**Presentation:** 3
**Contribution:** 1
**Rating:** 2
**Confidence:** 4

**Summary:**

The paper considers how to synthesize correct distributed protocols by combining Monte Carlo tree search, SMT-based correctness analysis, and specialized pruning procedures for guidance. Preliminary experiments and ablation studies are reported.

**Strengths:**

Applying learning for knowledge discovery is an exciting goal.

**Weaknesses:**

I do not find the motivation compelling at all. Distributed algorithms is a fascinating area where very smart people have contributed highly sophisticated algorithms over decades. It's unclear how the methods described in this paper can advance the area. Below are a few representative gaps between the toy setting used in the paper and distributed protocols: (1) use of FSMs to describe individual processes of a protocol is common, but one needs some form of symbolic descriptions that uses variables (of finite types) and guards and transitions over these. This is common in theory (e.g. see Nancy Lynch's 1995 book on Distributed Algorithms) and in protocol analysis tools (e.g. model checkers such as SPIN and NuSMV). (2) Process descriptions are symmetric but not identical (e.g. they make decisions based on comparison with Process ID). (3) Real challenges in protocol analysis are in implementation: high-level algorithms are typically published in a conference such as PODC with correctness proofs, and bugs show up in their translation to real-world platforms; contemporary work on protocol verification focuses on implementations (4) Protocols are typically analyzed using model checkers which can check both safety and liveness properties, SMT solvers are very limited since they can check only a small finite unrolling of the protocol (they are more scalable, but model checkers can easily analyze the toy state machines you are generating). In theory, one could  study the synthesis problem in a restrictive setting with the goal of removing these restrictions one by one, but the proposed approach does not inspire such confidence (for me).

**Questions:**

1. Why insist on "without human knowledge" and from scratch when this is an area with thousands of published papers?
2. If you are thinking a protocol that you would like to synthesize (and is not currently "known"), can you try asking a SOTA LLM (GPT-5) to produce it? It's not guaranteed to be correct, but can be checked by a model checker and counterexample can be used for prompting.

---

> ### Author Response · Authors · 2025-11-21
>
> **Weaknesses:**
> I do not find the motivation compelling at all. Distributed algorithms is a fascinating area where very smart people have contributed highly sophisticated algorithms over decades. It's unclear how the methods described in this paper can advance the area. Below are a few representative gaps between the toy setting used in the paper and distributed protocols: (1) use of FSMs to describe individual processes of a protocol is common, but one needs some form of symbolic descriptions that uses variables (of finite types) and guards and transitions over these. This is common in theory (e.g. see Nancy Lynch's 1995 book on Distributed Algorithms) and in protocol analysis tools (e.g. model checkers such as SPIN and NuSMV). (2) Process descriptions are symmetric but not identical (e.g. they make decisions based on comparison with Process ID). (3) Real challenges in protocol analysis are in implementation: high-level algorithms are typically published in a conference such as PODC with correctness proofs, and bugs show up in their translation to real-world platforms; contemporary work on protocol verification focuses on implementations (4) Protocols are typically analyzed using model checkers which can check both safety and liveness properties, SMT solvers are very limited since they can check only a small finite unrolling of the protocol (they are more scalable, but model checkers can easily analyze the toy state machines you are generating). In theory, one could study the synthesis problem in a restrictive setting with the goal of removing these restrictions one by one, but the proposed approach does not inspire such confidence (for me).
>
> **Response**: These are excellent points.
>
> (1) and (4) are somewhat correlated. We agree that using symbolic descriptions together with model checking is perhaps the most generic approach to describe a state machine. However, we do not know a way to learn those symbolic variables and transition rules. LLM may be the answer but it is not conclusive so far (see our experience with GPT as below). Our approach, which models the whole state as a number of bits, is equivalent to the symbolic approach in theory, since every variable eventually is represented as some bits in computers. SMT is sufficient when the number of processes is fixed. The limitation of our approach, which is acknowledged in the paper, is that it does not generalize to an arbitrary number of processes (model checking is useful in this case). Please see the \[Scalability\] discussion at the beginning.
>
> We don’t understand (2). Our work also incorporates process IDs, though for consensus and atomic commit, we find process ID is not necessary. We actually tried a third protocol, distributed locking, in which process ID is necessary, but since it is even simpler, we did not include it in the submission.
>
> For (3), implementation bugs are indeed something this work does not target. However, we want to emphasize that there are still many papers about protocol-level improvement and optimization every year, and our work targets automating or accelerating this line of work.
>
> **Questions:**
>
> 1. Why insist on "without human knowledge" and from scratch when this is an area with thousands of published papers?
> 2. If you are thinking a protocol that you would like to synthesize (and is not currently "known"), can you try asking a SOTA LLM (GPT-5) to produce it? It's not guaranteed to be correct, but can be checked by a model checker and counterexample can be used for prompting.
>
> **Response**:
>
> 1\. Our work is motivated by AlphaGo-Zero’s experience that zero-knowledge learning can outperform learning from human knowledge in certain areas and the similarity between distributed protocols and board games.
>
> 2\. This is an excellent suggestion. We tried GPT 5.1 with extended thinking and with the authors to manually provide counter examples. We document our experience in the evaluation and the details in appendix. In short, GPT finds a correct protocol for consensus but struggles with atomic commit. After we provide counterexamples for two iterations, GPT concludes that such a protocol is impossible. This is probably because there are existing protocols for consensus, but for atomic commit, since we change its problem definition to some extent, there are no existing protocols as far as we know.
>
> Of course this study is limited, with only two simple protocols on one LLM. It might be valuable to create a benchmark using questions related to distributed protocols and evaluate it on all major LLMs.

---

> > ### Comment · Reviewer_ms1M · 2025-11-24
> >
> > Thank you for a thoughtful response. Regarding my point (2) in the original review, maybe it's my own misunderstanding, so clarification will help. It seems that ProcessID is used to define global state. But since each FSM is identical, does it mean that its transitions cannot depend on its ProcessID. In particular, can we specify/synthesize a leader election protocol?

---

> > > ### Author Response · Authors · 2025-11-24
> > >
> > > Although FSM is identical, the incorporation of process ID as the input allows different processes to do different things. Concretely, an FSM may encode some logic like {if process ID = 0; do A; if process ID = 1; do B; ...}, so even if we deploy the same FSM to all processes, they can still do different things based on such logic. Our current implementation already supports this.
> > >
> > > Specifying a leader based protocol requires more than that. It should allow a process to not send messages (e.g., only the leader is broadcasting) or send messages to a subset of processes (e.g., a process replies to only the leader) in one round. Our current implementation does not support this and we discussed this in the future work.

---

> > > > ### Comment · Reviewer_ms1M · 2025-11-24
> > > >
> > > > Thanks, this answers my question.

---

### Author Response · Authors · 2025-11-21

Thanks for the insightful feedback. As all can see, this is exploratory work in a new area, certainly with many limitations. We regard this as grounding work, as reviewer EcFw pointed out, and plan to address its limitations in the future.

\[**Scalability**\] This is a common concern. We have added a discussion in Section 5\. In short, the scalability of our current implementation is indeed not very good. Engineering efforts may improve that, but the exponential growth pattern will probably still exist. However, we do not think scaling to a large number of processes is necessary to design a distributed protocol. Human experts often start from a small number of processes and then generalize the insights to an arbitrary number of processes. Therefore, finding a solution in small settings is still valuable. How to generalize is challenging and is our future work, but that does not require training a large-scale protocol.

---

> ### Author Response · Authors · 2025-12-03
>
> We have implemented early decision. We have finished the experiments on the setting con-3-2. Like other experiments, we ran this setting 10 times. The success rate is 100%. The running time is 214 minutes on average (min 177 minutes and max 251 minutes). This is longer than only allowing decision in the last round (117 minutes on average). We will continue to test other settings.

---

### Meta-Review · Area_Chair_54S9 · 2026-01-05

**Summary:**

Reviewers found the general topic of applying learning for knowledge discovery to be exciting, but are generally unconvinced by the approach taken in this paper. Some questions were clarified during the rebuttal (before the chaos happened) but without further discussions with the reviewers, who was originally very negative, it is unclear whether these will change the reviewer's sentiment. Overall, the sentiment was not strong enough to warrantee acceptance.

**Reviewer Concerns:**

see above

**Reviewer Scores:**

N/A

---

### Decision · Program_Chairs · 2026-01-26

Reject